# CELLVERSE: Do Large Language Models Really Understand Cell Biology?

**Fan Zhang[1], Tianyu Liu[2], Zhihong Zhu[3], Hao Wu[4], Haixin Wang[5],**
**Donghao Zhou[1], Yefeng Zheng[6], Kun Wang[7,†], Xian Wu[8,†], Pheng-Ann Heng[1,†]**

[1]CUHK, [2]Yale University, [3]Peking University, [4]Tsinghua University,
[5]UCLA, [6]Westlake University, [7]NTU, [8]Tencent
fzhang@link.cuhk.edu.hk, pheng@cse.cuhk.edu.hk
Project Page: https://cellverse-cuhk.github.io

## Abstract

Recent studies have demonstrated the feasibility of modeling single-cell data as natural languages and the potential of leveraging powerful large language models (LLMs) for understanding cell biology. However, a comprehensive evaluation of LLMs' performance on language-driven single-cell analysis tasks remains unexplored. Motivated by this challenge, we introduce CELLVERSE, a unified language-centric question-answering benchmark that integrates four types of single-cell multi-omics data and encompasses three hierarchical levels of single-cell analysis tasks: cell type annotation (cell-level), drug response prediction (drug-level), and perturbation analysis (gene-level). Going beyond this, we systematically evaluate the performance across 14 open-source and closed-source LLMs ranging 160M → 671B on CELLVERSE. Remarkably, the experimental results reveal: ① Existing specialist models (*e.g.*, `C2S-Pythia`) fail to make reasonable decisions across all sub-tasks within CELLVERSE, while generalist models such as `Qwen`, `Llama`, `GPT`, and `DeepSeek` family models exhibit preliminary understanding capabilities within the realm of cell biology. ② The performance of current LLMs falls short of expectations and has substantial room for improvement. Notably, in the widely studied drug response prediction task, none of the evaluated LLMs demonstrate significant performance improvement over *random guessing*. CELLVERSE offers the first large-scale empirical demonstration that significant challenges still remain in applying LLMs to cell biology. By introducing CELLVERSE, we lay the foundation for advancing cell biology through natural languages and hope this paradigm could facilitate next-generation single-cell analysis.

## 1 Introduction

Single-cell analysis [37, 32, 23, 60] has received growing attention in recent years due to its powerful capabilities across a wide range of healthcare applications, including disease diagnosis [9], drug discovery [21], and immunotherapy [27]. With the rapid progress in artificial intelligence and deep learning [31, 54], methodological advances in single-cell analysis have undergone a notable shift from traditional statistical techniques [29, 6] to specialized deep learning models [35, 41], and more recently, to approaches based on large-scale pre-trained foundation models [14, 20]. While these developments have led to significant improvements in performance and scalability, several inherent limitations remain unresolved (Figure 1): (1) *Lack of Unification.* For different types of omics data and downstream tasks, existing paradigms typically require separately designed models, lacking a unified approach capable of simultaneously handling multi-omics and multi-task scenarios.

---

[†]Corresponding authors.

39th Conference on Neural Information Processing Systems (NeurIPS 2025) Track on Datasets and Benchmarks.

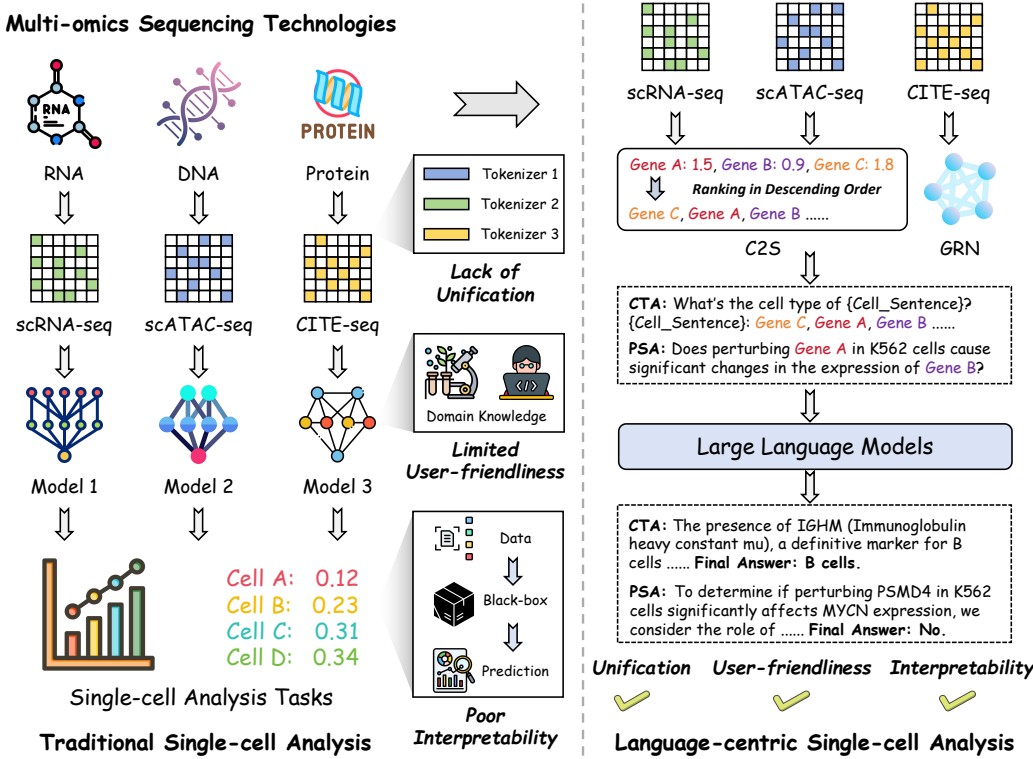

Figure 1: An illustration of traditional single-cell analysis and language-centric single-cell analysis.

(2) *Limited User-Friendliness.* Effective application of these methods to single-cell analysis often necessitates domain expertise in biology as well as proficiency in programming. Furthermore, the lack of user-centric interaction design in current models poses a significant barrier to adoption by non-expert users. (3) *Poor Interpretability.* Most of the existing data-driven black-box models directly learn the mapping from input (*e.g.*, gene expression) to output (*e.g.*, cell type information), without incorporating interpretable intermediate steps. As a result, users are often unable to understand the rationale behind the model's decisions. To this end, we seek to establish a unified, user-friendly, and interpretable paradigm for single-cell analysis. Leveraging the advanced techniques, such as cell2sentence [34] and gene regulatory network [28, 16], we can convert single-cell data into natural language formats. This transformation allows us to reformulate single-cell analysis tasks as question-answering (QA) problems—an interface that aligns with the capabilities of large language models (LLMs) and facilitates end-to-end reasoning over domain-specific knowledge in cell biology. With the reasoning capabilities and easy deployment of advanced LLMs, even non-expert users can efficiently analyze single-cell data, with interpretability supported via explicit querying of reasoning trajectories.

Specifically, we introduce CELLVERSE, a unified language-centric benchmark dataset for evaluating the capabilities of LLMs in single-cell analysis. We begin by curating five sub-datasets spanning four types of single-cell multi-omics data (scRNA-seq [51], CITE-seq [43], ASAP-seq [43], and scATAC-seq data [3]) and translate them into natural languages. Subsequently, we select three most representative single-cell analysis tasks—cell type annotation [48] (cell-level), drug response prediction [2] (drug-level), and perturbation analysis [25] (gene-level)—and reformulate them as QA problems by integrating each with the natural language-formatted single-cell data. Next, we conduct a comprehensive and systematic evaluation of 14 advanced LLMs on the proposed CELLVERSE benchmark. The evaluated models include open-source LLMs such as `C2S-Pythia (160M, 410M, and 1B)` [34, 50], `Qwen-2.5 (7B, 32B, and 72B)` [58], `Llama-3.3-70B` [18], and `DeepSeek (V3 and R1)` [38, 19], as well as closed-source models including `GPT-4` [1], `GPT-4o-mini` [44], `GPT-4o` [45], `GPT-4.1-mini` [46], and `GPT-4.1` [46].

Through a comprehensive analysis of the experimental results, we observe the following key findings: (1) Specialist models (`C2S-Pythia`), despite being trained specifically on single-cell analysis tasks, consistently exhibit hallucination issues across all tasks. Due to limited model capacity and

insufficient training data, they fail to make accurate decisions. In contrast, generalist models, though not fine-tuned for single-cell analysis, display initial reasoning capabilities and perform reasonably across various tasks. More importantly, some models not only produce task-specific predictions but also generate complete reasoning paths [56], demonstrating the potential of advanced LLMs for understanding cell biology. (2) However, generalist models still fall short of expectations. For the cell type annotation task, the state-of-the-art (SOTA) model achieves accuracies of $42.38\%$ on scRNA-seq data, $61.43\%$ on CITE-seq data, and $29.33\%$ on ASAP-seq data, indicating considerable room for improvement. On the more challenging tasks of drug response prediction and perturbation analysis (significance and direction), SOTA accuracies reach $55\%$, $76.67\%$, and $62.96\%$, respectively—most of them do not significantly outperform random guessing.

In summary, the main contributions of this paper are as follows:

- We identify key limitations in unification, user-friendliness, and interpretability of existing single-cell analysis paradigms. To address these issues, we propose a novel perspective: transforming single-cell data into natural languages and leveraging advanced LLMs for language-driven analysis.

- We propose CELLVERSE, a unified language-centric benchmark dataset for single-cell analysis that covers four types of single-cell multi-omics data and three representative sub-tasks. To the best of our knowledge, CELLVERSE is the first dataset designed to evaluate the understanding capabilities of LLMs in the domain of cell biology, and serves as a foundation for future research in this area.

- We conduct a comprehensive and systematic evaluation of 14 open-source and closed-source advanced LLMs on CELLVERSE, accompanied by in-depth analysis. Our analyses and findings offer insights and potential directions for future research in applying LLMs to cell biology.

## 2 Related Work

### 2.1 Large Language Models for Scientific Problems

Recent advances in large language models (LLMs) have spurred interest in applying them to scientific domains, including mathematics [24, 4], chemistry [47, 61], and biology [59, 42, 39]. For example, SciBERT [8] and BioBERT [33] leverage domain-specific pretraining on biomedical corpora to enhance their performance in biomedical text mining tasks, such as named entity recognition and relation extraction. Galactica [53] strives to unify scientific knowledge representation and generation, whereas AlphaCode [36] expands LLMs into program synthesis for scientific computation. These models exhibit promising early achievements in various tasks, including summarizing scientific literature, symbolic reasoning, and tackling textbook-style problems. However, most applications focus on general-purpose scientific texts [12] or structured symbolic inputs [57]. In contrast, the utilization of LLMs for domain-specific and data-intensive tasks, such as single-cell analysis, remains relatively unexplored. Our work contributes to this emerging field by connecting LLMs with scientific biological data, aiming to evaluate and improve LLM capabilities in real-world biological contexts.

### 2.2 Benchmark Datasets for Science QA

A variety of question-answering (QA) benchmark datasets have been proposed to evaluate the performance of advanced LLMs in scientific domains [52, 13]. Notable examples include SciBench [55], MathVista [40], and PubMedQA [26], which cover college-level scientific problems, multimodal mathematical reasoning problems, and biomedical literature, respectively. These benchmarks primarily focus on evaluating the capability of textual understanding and reasoning with easy-to-handle scientific contexts. However, none of the existing benchmarks are applicable to cell biology or account for the high-dimensional and sparse nature of single-cell data. Our work fills this gap by introducing a benchmark tailored to evaluating LLMs on biological QA tasks at the single-cell level.

## 3 The CELLVERSE Dataset

In this section, we will introduce the proposed CELLVERSE dataset. Firstly, in Section 3.1, we provide relevant preliminaries and background information. Then, in Section 3.2, we detail the dataset curation process. Finally, in Section 3.3, we present some key statistics of the dataset.

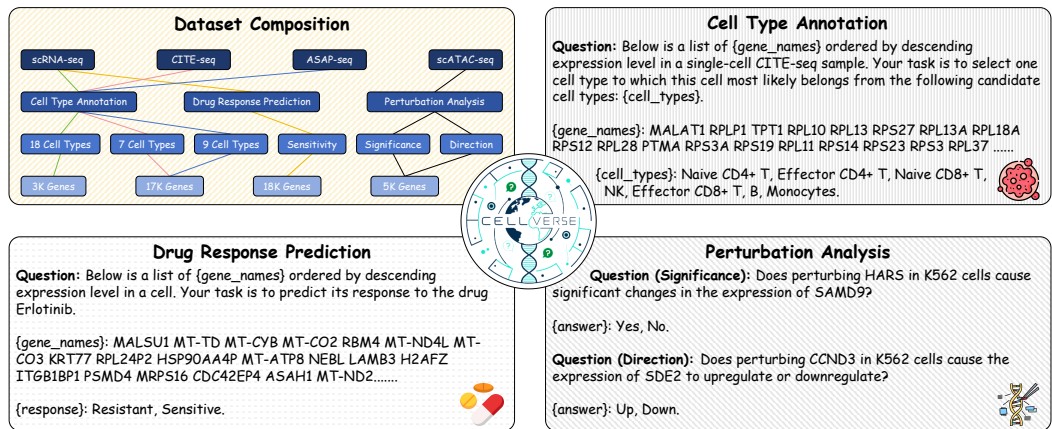

Figure 2: An overview of CELLVERSE. The top-left panel illustrates the composition of CELLVERSE, while the remaining three panels provide concrete data examples within CELLVERSE.

## 3.1 Preliminaries and Background

Single-cell data are characterized by long sequences [49], high sparsity [7], substantial noise [10], and strong heterogeneity [11]. Given a single-cell dataset $X \in \mathbb{R}^{N \times G}$, where $N$ denotes the number of cells and $G$ represents the number of features (typically corresponding to genes), the data structure varies across sequencing technologies and omics types. Depending on the modality, single-cell data may capture gene expression (scRNA-seq data), chromatin accessibility (scATAC-seq data), protein abundance (CITE-seq data), or DNA peak signals (ASAP-seq data). Due to the inherent input format and context length limitations of LLMs, efficiently transforming raw single-cell data into natural languages that retain key information with minimal loss remains a significant challenge. In the following sections, we introduce two feasible approaches to address this problem: (1) *cell2sentence (C2S)*, which encodes cell-level information into natural language, and (2) *gene regulatory network (GRN)*, which converts gene-level interactions into interpretable texts.

**Cell2sentence.** To transform single-cell data into natural languages, cell2sentence (C2S) [34] offers an intuitive perspective: it connects normalized expression profiles to natural languages by leveraging gene names and their ranked expression levels. Specifically, C2S treats gene names as tokens and represents each cell as a sentence composed of the top $n$ most highly expressed genes in descending order. This process can be formulated as:

$$\text{Cell Sentence}_i = [g_i^{(1)}, g_i^{(2)}, \ldots, g_i^{(n)}], \tag{1}$$

where $g_i^{(j)}$ denotes the gene name ranked $j$-th in expression level for cell $x_i \in \mathbb{R}^{1 \times G}$. This translation preserves essential cellular characteristics in a compact and interpretable form with controllable context length, facilitating the application of LLMs to high-dimensional single-cell data.

**Gene Regulatory Network.** To further enable natural language understanding of cell biology at the gene level, we leverage the gene regulatory network (GRN) [28, 16], which offers structured information of gene–gene interactions. A GRN is typically modeled as a directed graph $\mathcal{G} = (\mathcal{V}, \mathcal{E})$, where nodes $\mathcal{V}$ represent genes and edges $\mathcal{E}$ represent regulatory relationships inferred from expression data or external databases. Each edge $(g^a, g^b) \in \mathcal{E}$ with an associated weight is then translated into a natural language statement. Specifically, a nonzero edge weight implies that perturbing gene $g^a$ leads to a change in the expression of gene $g^b$. Conversely, if no edge exists or the weight is negligible, it suggests that perturbing $g^a$ does not significantly affect $g^b$. This process can be formulated as:

$$(g^a, g^b, w_{ab}) \in \mathcal{E}' \quad \Rightarrow \quad \begin{cases} \delta(g^a) \to \text{Change}(g^b), & \text{if } w_{ab} \geq \tau \\ \delta(g^a) \nrightarrow \text{Change}(g^b), & \text{if } w_{ab} < \tau \end{cases}, \tag{2}$$

where $w_{ab}$ is the edge weight indicating the regulatory strength from $g^a$ to $g^b$, $\delta$ is the perturbation operator, and $\tau$ is a predefined threshold to determine biological significance. This transformation

offers an interpretable summary of gene regulatory mechanisms, allowing LLMs to reason about biological pathways and perturbations. By expressing these regulatory dependencies in natural language, we effectively bridge graph-based biological knowledge and language-driven inference.

## 3.2 Data Curation

With the help of C2S and GRN, we can transform various single-cell analysis tasks into a unified QA format. Next, we detail the data curation process for CELLVERSE, which encompasses three core tasks: cell type annotation, drug response prediction, and perturbation analysis.

**Cell Type Annotation.** For the cell type annotation task, we first use C2S to convert raw single-cell data into cell sentences, and then directly query the corresponding cell type for each sentence. After generating the initial QA pairs, we perform two post-processing steps. First, we filter out low-quality samples with excessive redundancy based on sentence similarity. Second, we apply a resampling strategy to promote a more balanced data distribution across all cell types. An example of the resulting data for the cell type annotation task is shown in the top-right part of Figure 2.

**Drug Response Prediction.** For the drug response prediction task, we similarly use C2S to convert single-cell data into cell sentences, then query each sentence's response to a specific drug. After generating the QA pairs, we perform the same two post-processing steps as in the previous task: filtering out low-quality samples with high redundancy and promoting a more balanced data distribution. An example for drug response prediction is shown in the bottom-left panel of Figure 2.

**Perturbation Analysis.** For the perturbation analysis task, we leverage the GRN to identify interactions between genes. Specifically, for each candidate gene pair, we perform a non-parametric Wilcoxon test [17, 15] between the perturbed and control groups. An interaction is considered significant if the p-value is below 0.05 and the $\log_2$ fold change ($\log_2$FC) exceeds 0.5, indicating that perturbing gene $g^a$ induces a statistically significant change in the expression of gene $g^b$. Based on this criterion, we define two sub-tasks: (1) perturbation significance analysis, which asks whether perturbing gene $g^a$ significantly affects gene $g^b$; and (2) perturbation direction analysis, which further queries whether the expression of gene $g^b$ increases or decreases following the perturbation of gene $g^a$. To ensure data quality and coverage, we retain only samples with more than 10 cells in both perturbed and control groups and limit each source gene to a maximum of three QA examples. An example for the above two tasks is presented in the bottom-right panel of Figure 2.

## 3.3 Data Statistics

After completing the data curation process, we construct the CELLVERSE dataset, which integrates four types of single-cell multi-omics data and spans three sub-tasks (as illustrated in Figure 3). **Cell Type Annotation (CTA).** We include data from three different omics modalities: (1) scRNA-seq Multiple Sclerosis data [51]: This dataset contains 3,000 gene expression profiles and 18 annotated cell types: *phagocyte*, *cortical layer 2–3 excitatory neuron A*, *cortical layer 4 excitatory neuron*, *mixed glial cell*, *SV2C-expressing interneuron*, *microglial cell*, *cortical layer 5–6 excitatory neuron*, *oligodendrocyte A*, *cortical layer 2–3 excitatory neuron B*, *mixed excitatory neuron*, *endothelial cell*, *VIP-expressing interneuron*, *PVALB-expressing interneuron*, *oligodendrocyte precursor cell*, *pyramidal neuron*, *SST-expressing interneuron*, *astrocyte*, and *oligodendrocyte C*. (2) CITE-seq PBMC data [43]: This dataset includes 17,441 genes, annotated with 7 cell types: *Naive CD4+ T*, *Effector CD4+ T*, *Naive*

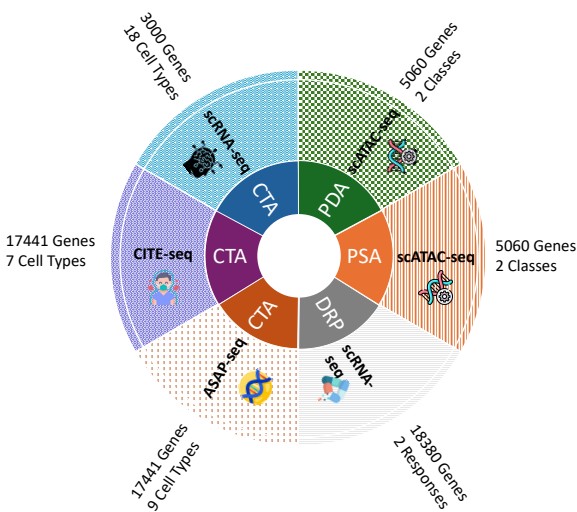

Figure 3: Data statistics information of CELLVERSE.

| Rank | Model | Score |
|------|-------|-------|
| 1 | DeepSeek-R1 | 42.38 |
| 2 | GPT-4.1-mini | 40.51 |
| 3 | GPT-4.1 | 37.83 |
| 4 | DeepSeek-V3 | 37.57 |
| 5 | GPT-4o | 35.70 |
| 6 | GPT-4 | 35.16 |
| 7 | Llama-3.3-70B | 32.75 |
| 8 | Qwen-2.5-72B | 24.73 |
| 9 | GPT-4o-mini | 23.93 |
| 10 | Qwen-2.5-32B | 22.46 |
| 11 | Qwen-2.5-7B | 13.77 |
| 12 | C2S-Pythia | 0 |

Cell Type Annotation
(scRNA-seq)

| Rank | Model | Score |
|------|-------|-------|
| 1 | GPT-4.1 | 61.43 |
| 2 | GPT-4.1-mini | 59.14 |
| 3 | GPT-4o | 58.29 |
| 4 | DeepSeek-R1 | 58.29 |
| 5 | DeepSeek-V3 | 57.14 |
| 6 | GPT-4 | 53.43 |
| 7 | Llama-3.3-70B | 52.57 |
| 8 | Qwen-2.5-72B | 50.29 |
| 9 | Qwen-2.5-32B | 48.86 |
| 10 | GPT-4o-mini | 48.57 |
| 11 | Qwen-2.5-7B | 30.86 |
| 12 | C2S-Pythia | 0 |

Cell Type Annotation
(CITE-seq)

| Rank | Model | Score |
|------|-------|-------|
| 1 | GPT-4.1-mini | 29.33 |
| 2 | Qwen-2.5-72B | 28.44 |
| 3 | GPT-4.1 | 28.22 |
| 4 | GPT-4o | 28.00 |
| 4 | DeepSeek-R1 | 28.00 |
| 6 | DeepSeek-V3 | 27.11 |
| 7 | Qwen-2.5-32B | 23.11 |
| 8 | Llama-3.3-70B | 22.00 |
| 9 | GPT-4 | 21.56 |
| 10 | GPT-4o-mini | 16.89 |
| 11 | Qwen-2.5-7B | 10.67 |
| 12 | C2S-Pythia | 0 |

Cell Type Annotation
(ASAP-seq)

| Rank | Model | Score |
|------|-------|-------|
| 1 | GPT-4.1-mini | 55.00 |
| 2 | DeepSeek-V3 | 50.63 |
| 3 | DeepSeek-R1 | 50.00 |
| 4 | Qwen-2.5-72B | 50.00 |
| 5 | Qwen-2.5-7B | 49.38 |
| 5 | GPT-4.1 | 49.38 |
| 7 | GPT-4o | 47.50 |
| 8 | Qwen-2.5-32B | 45.63 |
| 9 | GPT-4o-mini | 43.75 |
| 9 | Llama-3.3-70B | 43.75 |
| 11 | GPT-4 | 1.25 |
| 12 | C2S-Pythia | 0 |

Drug Response Prediction
(scRNA-seq)

| Rank | Model | Score |
|------|-------|-------|
| 1 | DeepSeek-R1 | 76.67 |
| 2 | DeepSeek-V3 | 76.67 |
| 3 | Qwen-2.5-32B | 76.67 |
| 4 | Qwen-2.5-7B | 76.67 |
| 5 | Qwen-2.5-72B | 73.33 |
| 5 | GPT-4.1 | 73.33 |
| 7 | GPT-4o | 71.67 |
| 8 | GPT-4.1-mini | 68.33 |
| 9 | Llama-3.3-70B | 66.67 |
| 10 | GPT-4o-mini | 41.67 |
| 11 | GPT-4 | 0 |
| 12 | C2S-Pythia | 0 |

Perturbation Significance Analysis
(scATAC-seq)

| Rank | Model | Score |
|------|-------|-------|
| 1 | DeepSeek-R1 | 62.96 |
| 2 | GPT-4.1-mini | 61.11 |
| 3 | Llama-3.3-70B | 57.41 |
| 4 | GPT-4o | 55.56 |
| 5 | GPT-4.1 | 46.30 |
| 6 | Qwen-2.5-72B | 35.19 |
| 7 | GPT-4o-mini | 31.48 |
| 8 | Qwen-2.5-7B | 29.63 |
| 9 | DeepSeek-V3 | 24.07 |
| 10 | Qwen-2.5-32B | 11.11 |
| 11 | GPT-4 | 0 |
| 12 | C2S-Pythia | 0 |

Perturbation Significance Analysis
(scATAC-seq)

Figure 4: Leaderboard results on our CELLVERSE benchmark. The scores represent the prediction accuracy of LLMs. We include both open-source and closed-source LLMs in the evaluation.

*CD8+ T*, *NK*, *Effector CD8+ T*, *B*, and *Monocytes*. (3) ASAP-seq PBMC data [43]: This dataset also includes 17,441 genes and 9 cell types: *DC*, *Naive CD4+ T*, *Effector CD4+ T*, *Naive CD8+ T*, *NK*, *Effector CD8+ T*, *B*, *Monocytes*, and *unknown*. **Drug Response Prediction (DRP).** We utilize a scRNA-seq dataset [5] measuring cellular responses to the drug erlotinib. This dataset contains 18,380 genes and two drug responses: *sensitive* and *resistant*. **Perturbation Analysis (PA).** For this task, we utilize scATAC-seq data from K562 cells [3], encompassing 5,060 genes, to define two sub-tasks: **Perturbation Significance Analysis (PSA).** A binary classification task that asks whether perturbing gene $g^a$ significantly affects gene $g^b$, with answer choices *yes* or *no*. **Perturbation Direction Analysis (PDA).** Another binary classification task that determines the direction of change for gene $g^b$ after perturbation of gene $g^a$, with answer choices *up* or *down*.

## 4 Experiments

### 4.1 Evaluation Protocols and Implementation Details

To evaluate the performance of current advanced LLMs on the proposed CELLVERSE, we conduct a fair comparison across 9 open-source models (C2S-Pythia-160M [34], C2S-Pythia-410M [34], C2S-Pythia-1B [50], Qwen-2.5-7B [58], Qwen-2.5-32B [58], Qwen-2.5-72B [58], Llama-3.3-70B [18], DeepSeek-V3 [38], and DeepSeek-R1 [19]) and 5 closed-source models (GPT-4 [1], GPT-4o-mini [44], GPT-4o [45], GPT-4.1-mini [46], and GPT-4.1 [46]). Among these, only the C2S-Pythia series models were trained specifically on single-cell analysis tasks and thus considered specialist models, while all others are generalist models. We perform inference using the vLLM framework [30] for open-source models and official APIs for closed-source models. During preliminary testing, we observed that all models struggle to make reasonable decisions under open-ended question settings. Therefore, we convert all questions into multiple-choice format by including a list of candidate answers directly in the prompt. The experiments are conducted

Table 1: Performance comparison (%) of cell type annotation on single-cell multi-omics data.

| Data | scRNA-seq | | | | CITE-seq | | | | ASAP-seq | | | |
|---|---|---|---|---|---|---|---|---|---|---|---|---|
| Metric | Precision | Recall | F1 | Accuracy | Precision | Recall | F1 | Accuracy | Precision | Recall | F1 | Accuracy |
| *Open-source LLMs* | | | | | | | | | | | | |
| Qwen-2.5-7B | 36.31 | 14.54 | 11.53 | 13.77 | 41.42 | 27.00 | 28.08 | 30.86 | 15.47 | 9.60 | 7.37 | 10.67 |
| Qwen-2.5-32B | 28.81 | 20.60 | 18.76 | 22.46 | 45.58 | 42.75 | 38.57 | 48.86 | 25.87 | 20.80 | 19.87 | 23.11 |
| Qwen-2.5-72B | 30.94 | 22.39 | 19.32 | 24.73 | 43.60 | 44.00 | 40.50 | 50.29 | 25.04 | 25.60 | 22.61 | 28.44 |
| Llama-3.3-70B | 33.60 | 30.25 | 24.99 | 32.75 | 61.30 | 52.57 | 50.02 | 52.57 | 26.95 | 22.00 | 19.30 | 22.00 |
| DeepSeek-V3 | 38.69 | 34.99 | 30.60 | 37.57 | 66.44 | 57.14 | 54.65 | 57.14 | 26.52 | 24.40 | 21.18 | 27.11 |
| DeepSeek-R1 | 39.95 | 38.81 | 33.40 | 42.38 | 57.57 | 51.00 | 50.76 | 58.29 | 32.69 | 25.20 | 20.81 | 28.00 |
| *Closed-source LLMs* | | | | | | | | | | | | |
| GPT-4 | 40.81 | 32.41 | 29.89 | 35.16 | 47.21 | 46.75 | 43.44 | 53.43 | 26.48 | 19.40 | 17.16 | 21.56 |
| GPT-4o-mini | 31.86 | 22.64 | 15.77 | 23.93 | 63.94 | 48.57 | 47.89 | 48.57 | 33.27 | 16.89 | 13.52 | 16.89 |
| GPT-4o | 40.24 | 31.52 | 29.03 | 35.70 | 60.07 | 58.29 | 55.76 | 58.29 | 32.47 | 28.00 | 25.00 | 28.00 |
| GPT-4.1-mini | 41.14 | 36.46 | 34.55 | 40.51 | 66.13 | 59.14 | 58.47 | 59.14 | 36.86 | 29.33 | 27.47 | 29.33 |
| GPT-4.1 | 42.46 | 35.19 | 30.94 | 37.83 | 68.32 | 61.43 | 59.26 | 61.43 | 42.53 | 28.22 | 23.99 | 28.22 |

Table 2: Performance comparison (%) of drug response prediction on scRNA-seq data.

| Setting | Overall | | | | Sensitive | | | Resistant | | |
|---|---|---|---|---|---|---|---|---|---|---|
| Metric | Precision | Recall | F1 | Accuracy | Precision | Recall | F1 | Precision | Recall | F1 |
| *Open-source LLMs* | | | | | | | | | | |
| Qwen-2.5-7B | 33.82 | 32.92 | 32.38 | 49.38 | 49.52 | 65.00 | 56.22 | 51.92 | 33.75 | 40.91 |
| Qwen-2.5-32B | 44.85 | 45.63 | 43.50 | 45.63 | 42.86 | 26.25 | 32.56 | 46.85 | 65.00 | 54.45 |
| Qwen-2.5-72B | 34.02 | 33.33 | 30.84 | 50.00 | 50.00 | 21.25 | 29.82 | 52.07 | 78.75 | 62.69 |
| Llama-3.3-70B | 43.71 | 43.75 | 43.67 | 43.75 | 44.19 | 47.50 | 45.78 | 43.24 | 40.00 | 41.56 |
| DeepSeek-V3 | 52.44 | 50.63 | 39.34 | 50.63 | 54.55 | 7.50 | 13.19 | 50.34 | 93.75 | 65.50 |
| DeepSeek-R1 | 25.00 | 50.00 | 33.33 | 50.00 | 0.00 | 0.00 | 0.00 | 50.00 | 100.00 | 66.67 |
| *Closed-source LLMs* | | | | | | | | | | |
| GPT-4 | 13.33 | 0.83 | 1.57 | 1.25 | 0.00 | 0.00 | 0.00 | 40.00 | 0.25 | 4.71 |
| GPT-4o-mini | 36.82 | 43.75 | 35.24 | 43.75 | 27.27 | 7.50 | 11.76 | 46.38 | 80.00 | 58.72 |
| GPT-4o | 24.78 | 31.67 | 22.96 | 47.50 | 25.00 | 2.50 | 4.55 | 49.33 | 92.50 | 64.35 |
| GPT-4.1-mini | 60.54 | 55.00 | 48.19 | 55.00 | 68.18 | 18.75 | 29.41 | 52.90 | 91.25 | 66.97 |
| GPT-4.1 | 44.84 | 49.38 | 35.12 | 49.38 | 40.00 | 2.50 | 4.71 | 49.68 | 96.25 | 65.53 |

under both zero-shot and few-shot settings. Model performance is then evaluated using standard metrics: precision score, recall score, F1 score, and overall accuracy.

## 4.2 Experimental Results and In-depth Analysis

In Figure 4, we present the leaderboard results for all sub-tasks. Since C2S-Pythia-160M, C2S-Pythia-410M, and C2S-Pythia-1B perform similarly across all tasks, we aggregate their results and refer to them collectively as C2S-Pythia. Detailed results for each sub-task—cell type annotation, drug response prediction, perturbation significance analysis, and perturbation direction analysis—are provided in Table 1, Table 2, Table 3 3, and Table 4, respectively. The best and second-best results in the tables are marked in red and blue, respectively. More results and analysis can be found in Appendix B, C, and D. From the results, we can derive the following observations.

**Obs. 1: Generalist Models Perform Better than Specialist Models.** Although the specialist models have been trained on language-centric single-cell analysis tasks, experimental results in Figure 4 show that C2S-Pythia fails to produce reliable predictions across all sub-tasks on CELLVERSE. This suggests that, due to limitations in model capacity and training data, the specialist models may be overfitting rather than learning to generalize to unseen questions. In contrast, despite lacking task-specific training, larger-capacity generalist models exhibit emerging capabilities in reasoning about cell biology. These findings highlight the potential of leveraging strong generalist models as base architectures for future research in language-driven single-cell analysis.

**Obs. 2: LLM Performance Scales with Model Size.** We observe that GPT-4 and DeepSeek family models generally outperform the Llama and Qwen series across all tasks. This trend suggests a positive correlation between model capacity and performance in cell biology understanding. Notably, across the six task-specific leaderboards, all top-performing models belong to either the DeepSeek or GPT-4 families—securing four and two first-place rankings, respectively. These results indicate that the scaling laws of LLMs also hold in the context of cell biology.

Table 3: Performance comparison (%) of perturbation significance analysis on scATAC-seq data.

| Setting | Overall | | | | Yes | | | No | | |
|---|---|---|---|---|---|---|---|---|---|---|
| Metric | Precision | Recall | F1 | Accuracy | Precision | Recall | F1 | Precision | Recall | F1 |
| *Open-source LLMs* | | | | | | | | | | |
| Qwen-2.5-7B | 38.33 | 50.00 | 43.40 | 76.67 | 0.00 | 0.00 | 0.00 | 76.67 | 100.00 | 86.79 |
| Qwen-2.5-32B | 38.33 | 50.00 | 43.40 | 76.67 | 0.00 | 0.00 | 0.00 | 76.67 | 100.00 | 86.79 |
| Qwen-2.5-72B | 37.93 | 47.83 | 42.31 | 73.33 | 0.00 | 0.00 | 0.00 | 75.86 | 95.65 | 84.62 |
| Llama-3.3-70B | 58.75 | 60.87 | 58.96 | 66.67 | 35.00 | 50.00 | 41.18 | 82.50 | 71.74 | 76.74 |
| DeepSeek-V3 | 63.79 | 52.48 | 49.52 | 76.67 | 50.00 | 7.14 | 12.50 | 77.59 | 97.83 | 86.54 |
| DeepSeek-R1 | 64.81 | 57.45 | 58.00 | 76.67 | 50.00 | 21.43 | 30.00 | 79.63 | 93.48 | 86.00 |
| *Closed-source LLMs* | | | | | | | | | | |
| GPT-4 | 0.00 | 0.00 | 0.00 | 0.00 | 0.00 | 0.00 | 0.00 | 0.00 | 0.00 | 0.00 |
| GPT-4o-mini | 53.97 | 54.50 | 41.52 | 41.67 | 25.58 | 78.57 | 38.60 | 82.35 | 30.43 | 44.44 |
| GPT-4o | 61.11 | 61.65 | 61.35 | 71.67 | 40.00 | 42.86 | 41.38 | 82.22 | 80.43 | 81.32 |
| GPT-4.1-mini | 25.79 | 29.71 | 27.61 | 68.33 | 0.00 | 0.00 | 0.00 | 77.36 | 89.13 | 82.83 |
| GPT-4.1 | 58.17 | 55.28 | 55.47 | 73.33 | 37.50 | 21.43 | 27.27 | 78.85 | 89.13 | 83.67 |

Table 4: Performance comparison (%) of perturbation direction analysis on scATAC-seq data.

| Setting | Overall | | | | Up | | | Down | | |
|---|---|---|---|---|---|---|---|---|---|---|
| Metric | Precision | Recall | F1 | Accuracy | Precision | Recall | F1 | Precision | Recall | F1 |
| *Open-source LLMs* | | | | | | | | | | |
| Qwen-2.5-7B | 60.78 | 17.06 | 24.36 | 29.63 | 82.35 | 41.18 | 54.90 | 100.00 | 10.00 | 18.18 |
| Qwen-2.5-32B | 27.08 | 9.31 | 11.11 | 11.11 | 50.00 | 2.94 | 5.56 | 31.25 | 25.00 | 27.78 |
| Qwen-2.5-72B | 38.10 | 25.49 | 29.17 | 35.19 | 64.29 | 26.47 | 37.50 | 50.00 | 50.00 | 50.00 |
| Llama-3.3-70B | 55.61 | 55.88 | 55.56 | 57.41 | 67.74 | 61.76 | 64.62 | 43.48 | 50.00 | 46.51 |
| DeepSeek-V3 | 35.95 | 16.86 | 22.94 | 24.07 | 41.18 | 20.59 | 27.45 | 66.67 | 30.00 | 41.38 |
| DeepSeek-R1 | 58.75 | 57.21 | 57.07 | 62.96 | 67.50 | 79.41 | 72.97 | 50.00 | 35.00 | 41.18 |
| *Closed-source LLMs* | | | | | | | | | | |
| GPT-4 | 0.00 | 0.00 | 0.00 | 0.00 | 0.00 | 0.00 | 0.00 | 0.00 | 0.00 | 0.00 |
| GPT-4o-mini | 22.82 | 24.90 | 20.43 | 31.48 | 38.46 | 14.71 | 21.28 | 30.00 | 60.00 | 40.00 |
| GPT-4o | 55.08 | 55.44 | 54.56 | 55.56 | 67.86 | 55.88 | 61.29 | 42.31 | 55.00 | 47.83 |
| GPT-4.1-mini | 43.18 | 40.59 | 41.63 | 61.11 | 75.00 | 61.76 | 67.74 | 54.55 | 60.00 | 57.14 |
| GPT-4.1 | 49.13 | 49.12 | 46.28 | 46.30 | 61.90 | 38.24 | 47.27 | 36.36 | 60.00 | 45.28 |

**Obs. 3: Current LLMs Demonstrate Limited Understanding of Cell Biology.** While LLMs exhibit preliminary capabilities for cell biology, their overall performance remains far from satisfactory. On the cell type annotation task, the top-performing models achieved accuracies of only 42.38%, 61.43%, and 29.33% on scRNA-seq, CITE-seq, and ASAP-seq data, respectively. For the binary drug response prediction task, the best model achieved only 55% accuracy, indicating no substantial improvement over random guessing. In the perturbation analysis tasks, we observe that LLMs perform better in assessing perturbation significance than in predicting perturbation direction, which aligns with intuitive expectations. Most models also fail to outperform random guessing in the direction prediction task. GPT-4 often refuses to answer for both drug response prediction and perturbation analysis tasks, while Qwen family models tend to answer "No" to all questions in the perturbation significance analysis task. Overall, these results highlight that current LLMs are far from satisfactory and there remains substantial room for improvement in LLMs for cell biology.

**Obs. 4: Scaling Context Lengths May Not Consistently Improve Performance.** Figure 5 presents the cell type annotation results of four advanced LLMs on CITE-seq data as the context length increases. We scale the context length by including more gene names in the cell sentences. For the GPT-4 family, performance generally improves with longer context lengths, suggesting these models benefit from richer input information. In contrast, for the DeepSeek family, increasing context length does not consistently lead to performance gains. We hypothesize that this is because DeepSeek models already demonstrate strong reasoning capabilities with relatively short contexts, and adding more genes with low expressions may introduce noise that hinders model performance.

**Obs. 5: Few-shot In-context Learning Does Not Always Boost Performance.** Unlike general tasks, few-shot in-context learning yields limited gains and often even degrades performance in

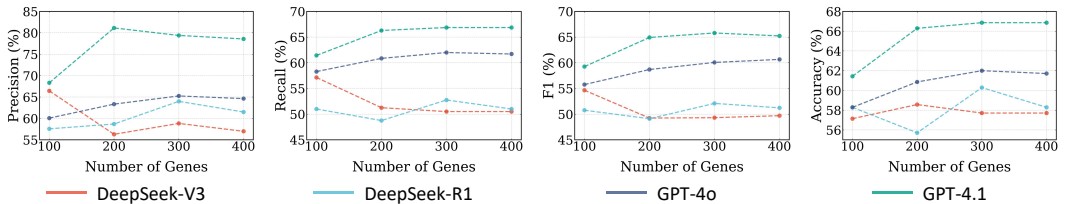

Figure 5: Performance comparison of cell type annotation when scaling context lengths.

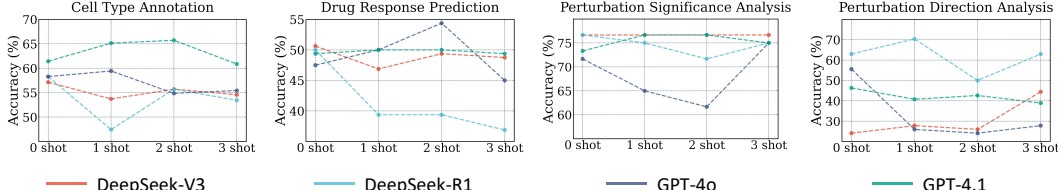

Figure 6: Performance comparison under few-shot settings across various tasks.

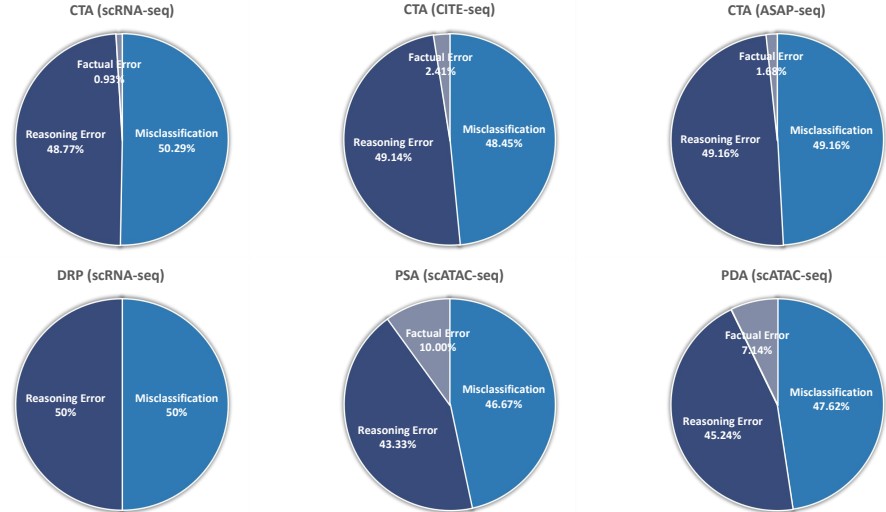

Figure 7: Distribution of `DeepSeek-R1`'s errors within distinct types across various tasks.

single-cell analysis. Results in Figure 6 exhibit no consistent improvement. We hypothesize that this is largely due to the high level of noise inherent in single-cell data, which hampers the model's ability to generalize when guided by noisy examples. This observation highlights an important insight: In few-shot settings, sample quality may be more critical than sample quantity.

**Obs. 6: Error Profiles Vary across Tasks.** Figure 7 shows the distribution of error types made by `DeepSeek-R1` across different tasks. We observe that the most frequent error categories are reasoning errors and misclassifications. Additionally, another type of error, factual errors, is more prominent in gene-level tasks, whereas they are relatively rare in cell-level and drug-level tasks.

## 5  Conclusion

In this work, we introduced CELLVERSE, a unified language-centric benchmark that encompasses single-cell multi-omics data and spans hierarchical single-cell analysis tasks, addressing the absence of systematic evaluation in LLMs for cell biology. Through a comprehensive assessment of 14 advanced LLMs, we uncovered both their capabilities and limitations in reasoning over single-cell analysis tasks, laying the groundwork for future research within this domain. In future works, we plan to (1) enhance the scalability and diversity of CELLVERSE, and (2) build on CELLVERSE to advance natural language understanding as a next-generation paradigm for interpreting cell biology.

## Acknowledgment

The work described in this paper was supported in part by the Research Grants Council of the Hong Kong Special Administrative Region, China, under Project T45-401/22-N; and under Project STG1/E-401/23-N.

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

# Appendix Overview

# A More Details about CELLVERSE

## A.1 Model Sources

For all evaluated LLMs, we select their latest versions and best-performing configurations to accurately assess their capabilities in understanding cell biology. Table 5 summarizes the release dates and sources of the models included in CELLVERSE.

Table 5: The release dates and sources of the evaluated models in CELLVERSE.

| Model | Release Date | Source |
|---|---|---|
| C2S-Pythia-160M[34] | 2024-02 | https://huggingface.co/vandijklab/pythia-160m-c2s |
| C2S-Pythia-410M [34] | 2024-09 | https://huggingface.co/vandijklab/C2S-Pythia-410m-diverse-single-and-multi-cell-tasks |
| C2S-Pythia-1B [50] | 2025-04 | https://huggingface.co/vandijklab/C2S-Scale-Pythia-1b-pt |
| Qwen-2.5-7B[58] | 2024-09 | https://huggingface.co/Qwen/Qwen2.5-7B |
| Qwen-2.5-32B[58] | 2024-09 | https://huggingface.co/Qwen/Qwen2.5-32B |
| Qwen-2.5-72B[58] | 2024-09 | https://huggingface.co/Qwen/Qwen2.5-72B |
| Llama-3.3-70B[18] | 2024-12 | https://huggingface.co/meta-llama/Llama-3.3-70B-Instruct |
| DeepSeek-V3 [38] | 2025-03 | https://huggingface.co/deepseek-ai/DeepSeek-V3 |
| DeepSeek-R1[19] | 2025-03 | https://huggingface.co/deepseek-ai/DeepSeek-R1 |
| GPT-4 [1] | 2023-06 | https://platform.openai.com/docs/models/gpt-4 |
| GPT-4o-mini [44] | 2024-07 | https://platform.openai.com/docs/models/gpt-4o-mini |
| GPT-4o [45] | 2024-11 | https://platform.openai.com/docs/models/gpt-4o |
| GPT-4.1-mini [46] | 2025-04 | https://platform.openai.com/docs/models/gpt-4.1-mini |
| GPT-4.1 [46] | 2025-04 | https://platform.openai.com/docs/models/gpt-4.1 |

## A.2 Evaluation Prompts for Single-cell Analysis Tasks

We design task-specific prompts to evaluate the performance of LLMs on different single-cell analysis tasks within CELLVERSE. At the beginning of each interaction, we add a unified system prompt instructing the LLMs to act as experts in cell biology and genomics. We then integrate the transformed language-centric information with the task-specific prompts to form the final questions. Since current LLMs struggle to produce accurate predictions in open-ended formats, we convert all questions into a closed-set, multiple-choice setting to ensure more reliable evaluation. As shown in Table 6, we summarize the prompt design strategies used for each task.

# B Additional Experimental Results

## B.1 Performance of Strengthening Open-source Models

As shown in Table 7, we include three more cutting-edge open-source LLMs for performance comparison, to alleviate the concerns of closed-source model dependency.

Table 6: Evaluation prompt of LLMs for different single-cell analysis tasks.

| Task | Prompt |
|---|---|
| Cell Type Annotation | You are an expert who knows a lot about single cell biology and genomics and will help me solve a series of tasks related to single cell data analysis. Below is a list of {gene_names} ordered by descending expression level in a single-cell sample. Your task is to select one cell type to which this cell most likely belongs from the following candidate cell types: {cell_types}. Make your choice in format 'Final Answer: Prediction'.
– **Gene Names:** {gene_names}
– **Cell Types:** {cell_types} |
| Drug Response Prediction | You are an expert who knows a lot about single cell biology and genomics and will help me solve a series of tasks related to single cell data analysis. Below is a list of {gene_names} ordered by descending expression level in a cell. Your task is to predict its response to the drug {drug}: Responses: {responses}. Make your choice in format 'Final Answer: Response'.
– **Gene Names:** {gene_names}
– **Drug:** {drug}
– **Responses:** {responses} |
| Perturbation Significance Analysis | You are an expert who knows a lot about single cell biology and genomics and will help me solve a series of tasks related to single cell data analysis. Does perturbing {gene_a} in K562 cells cause significant changes in the expression of {gene_b}? Make your choice in format 'Final Answer: Yes' or 'Final Answer: No'.
– **Gene A:** {gene_a}
– **Gene B:** {gene_b} |
| Perturbation Direction Analysis | You are an expert who knows a lot about single cell biology and genomics and will help me solve a series of tasks related to single cell data analysis. Does perturbing {gene_a} in K562 cells cause the expression of {gene_b} to upregulate or downregulate? Make your answer in format 'Final Answer: Up' or 'Final Answer: Down'.
– **Gene A:** {gene_a}
– **Gene B:** {gene_b} |

## B.2 Performance of Random Baseline

In Table 8, we add the performance of the random guess baseline to make the comparison more visually immediate.

## B.3 Performance Comparison of Specific Cell Types

We present fine-grained comparisons of model performance across specific cell types in the cell type annotation task. Figures 8, 9, and 10 report precision, recall, and F1 scores on scRNA-seq data, respectively. Similarly, Figures 11 and 12 show results on CITE-seq and ASAP-seq data. From the results, it can be observed that all the evaluated LLMs consistently struggle to identify certain challenging cell types, such as oligodendrocyte C and phagocyte.

Table 7: Cell type annotation results (%) of additional open-source models.

| Data | scRNA-seq | | | | CITE-seq | | | | ASAP-seq | | | |
|---|---|---|---|---|---|---|---|---|---|---|---|---|
| Metrics | Precision | Recall | F1 | Acc | Precision | Recall | F1 | Acc | Precision | Recall | F1 | Acc |
| LLaMA-3.1-8B | 21.54 | 11.82 | 8.16 | 11.9 | 35.49 | 20.5 | 22.02 | 23.43 | 9.7 | 2.6 | 3.93 | 2.89 |
| Qwen-3-8B | 32.15 | 24.9 | 19.57 | 25.27 | 44.68 | 39.5 | 39.98 | 45.14 | 24.7 | 18.2 | 16.06 | 20.22 |
| Qwen-3-14B | 41.41 | 28.72 | 27.59 | 32.22 | 55.44 | 52.29 | 50.4 | 52.29 | 25.77 | 21 | 19.85 | 23.33 |

Table 8: Performance (%) of the random guess baseline on various tasks.

| Task | CTA (scRNA-seq) | CTA (CITE-seq) | CTA (ASAP-seq) | DRP | PSA | PDA |
|---|---|---|---|---|---|---|
| Precision | 5.08 | 15.19 | 11.53 | 49.37 | 51.01 | 44.92 |
| Recall | 4.29 | 14.86 | 11.56 | 49.37 | 51.40 | 44.56 |
| F1 | 4.61 | 14.96 | 11.51 | 49.36 | 45.79 | 43.75 |
| Acc | 5.08 | 14.86 | 11.56 | 49.38 | 48.33 | 44.44 |

## B.4 More Metrics Comparison under Few-shot Settings

Figure 13 presents experimental results for precision, recall, and F1 score under few-shot settings across different single-cell analysis tasks. The results indicate that for all of the evaluated LLMs, increasing the number of few-shot samples does not necessarily lead to performance gains and can even degrade model performance, which is consistent with our earlier analysis. These findings suggest that in future research on LLMs for cell biology, selecting high-quality and representative few-shot samples will be a critical and challenging problem.

## C Additional Error Analysis

In Figure 14 and Figure 15, we additionally present the error type distributions of GPT-4.1 and GPT-4o across various tasks in CELLVERSE. The results show that, similar to the error type distribution of DeepSeek-R1, misclassification and reasoning errors are also the two most common error types for GPT-4.1 and GPT-4o. For cell-level tasks, factual errors are relatively rare, whereas their frequency increases in drug-level and gene-level tasks. This trend is expected, as drug-level and gene-level tasks are inherently more complex than cell-level ones.

We also include a discussion on in-depth error analysis about specific rare cell types:

---

**Cell Type Annotation**

**Question:** Below is a list of {gene_names} ordered by descending expression level in a single-cell CITE-seq sample. Your task is to select one cell type to which this cell most likely belongs from the following candidate cell types: {cell_types}.
{gene_names}: MALAT1 TMSB4X TPT1 RPL13 RPL10 RPS19 ACTB RPS12 RPS14 IL32 RPS3A LTB RPL30 RPLP1 RPL28 RPL13A RPS27 RPS3 RPS7 RPL3 HLA-B RPL14 RPS13 UBA52 RPS6 RPS23 RPS9 RPL11 HLA-C RPS15A RPS24 RPL6 RPL36 RPL37A HLA-A RPS21 RPS15 S100A4 RPL39 CYBA RPL29 RPL18A S100A6 PTPRC RPL22 PTMA NACA HLA-E RPS20 RPS29 RPL27 RPL9 EIF1 SERF2 CFL1 RPL38 HSP90AB1 RPL23A STAT1 RPL37 IFITM2 CALM1 ATP5F1E KIAA1551 ISG15 SON HLA-F PABPC1 CD48 YWHAZ PPIA RPL10A RPL5 HNRNPA2B1 STK17A HINT1 CALR UBB FTH1 RPSA GAPDH CORO1A PSME1 GIMAP7 XAF1 ZNF217 FXYD5 CD2 GADD45GIP1 NCL CYSLTR1 RPL24 RPL31 SAMD9 TRGC2 ENO1 TAF15 CDC42 PSMA7 SSB...
{cell_types}: Naive CD4+ T, Effector CD4+ T, Naive CD8+ T, NK, Effector CD8+ T, B, Monocytes.
**Ground Truth:** Effector CD8+ T

DeepSeek-R1: The gene list provided includes markers indicative of T cell identity (e.g., IL32, LTB, PTPRC) and lacks classical markers for B cells (e.g., CD79A, MS4A1), monocytes (e.g., CD14, LYZ), or NK cells (e.g., NKG7, GNLY). Among T cell subsets, IL32 is strongly associated with activated/effector CD4+ T cells, while the absence of cytotoxic genes (e.g., GZMB, PRF1) and CD8-specific markers (e.g., CD8A/B) makes CD8+ effector T cells less likely. Additionally, S100A4 and STAT1 suggest an activated/effector phenotype, further supporting effector CD4+ T cells over naive subsets (which typically express CCR7 or SELL).
**Final Answer:** Effector CD4+ T ✗

---

**Error Analysis.** The model incorrectly predicted "Effector CD4+ T" instead of the ground truth "Effector CD8+ T" due to:

1. Biased interpretation of IL32 as CD4+-specific,

2. Inadequate handling of single-cell data limitations (dropout effects),

3. Failure to recognize CD8+-specific signatures in the expression profile.

# D   Case Study

Here, we present a case study showcasing responses from different LLMs across various tasks in CELLVERSE, demonstrating their reasoning capabilities in cell biology.

---

**Cell Type Annotation**

**Question:** Below is a list of {gene_names} ordered by descending expression level in a multiple sclerosis cell. Your task is to select one cell type to which this cell most likely belongs from the following candidate cell types: {cell_types}.
{gene_names}: MT-CO2 ATP1B1 MT-CO3 MT-CO1 YWHAG RAB3B GAD1 MAP1B PSAP......
{cell_types}: phagocyte, cortical layer 2-3 excitatory neuron A, cortical layer 4 excitatory neuron......
**Ground Truth:** PVALB-expressing interneuron

`DeepSeek-R1`: The cell exhibits high expression of genes associated with GABAergic interneurons, including GAD1 and GAD2, which are critical for GABA synthesis. Synaptic markers like SNAP25, SYT1, and VAMP2 further confirm neuronal identity. Notably, KCNC2 (encoding Kv3.2, a potassium channel linked to fast-spiking interneurons) is present, which is characteristic of PVALB-expressing interneurons. While PVALB itself is not explicitly listed, the combination of GABAergic markers and KCNC2 strongly supports this subtype. Other interneuron subtypes (e.g., SST, VIP) lack their defining markers in the gene list. Glial or oligodendrocyte markers are absent, ruling out non-neuronal types.
**Final Answer:** PVALB-expressing interneuron ✓

`GPT-4.1`: Reasoning: The gene list is highly enriched for neuronal markers (e.g., SNAP25, SYT1, GAD1, GAD2, ERBB4, CALM3, ELAVL2, NDRG4, ATP1A3, KCNC2, DNER, PNMA2, ATP2A2, RAB3A, UCHL1, PRKACB, PRNP, RTN1, CPLX1, VAMP2, SLC24A2, STXBP1, GPM6A, SYN2, FGF12, DNM3, SLC6A1, SLC6A17, SYN1, SCAMP5, GABRA1, GABRG1, GLRA2, GLRA3, GRIK1, GRIK3, GABARAPL1, GABARAPL2, etc.). There is strong expression of GAD1 and GAD2, which are canonical markers of GABAergic interneurons. SV2C is not directly listed, but the presence of many synaptic vesicle and GABAergic genes (e.g., RAB3A/B/C, SNAP25, SYT1, STXBP1, VAMP2, SLC6A1, SLC6A17) supports an interneuron identity. There is no strong evidence for astrocyte, oligodendrocyte, microglial, or endothelial markers. Among interneuron subtypes, there is no strong expression of VIP, SST, or PVALB, which would suggest those specific subtypes. The overall profile, with high GABAergic and synaptic gene expression, best matches the SV2C-expressing interneuron among the provided options.
**Final Answer:** SV2C-expressing interneuron ✗

`GPT-4o`: To determine the most likely cell type based on the gene expression profile provided, we need to identify key marker genes that are characteristic of specific cell types. 1. Neuronal Markers: Genes such as SNAP25, SYT1, MAP1B, and GAD1 are typically associated with neurons. GAD1 and GAD2 are markers for GABAergic neurons, which include interneurons like SST-expressing, VIP-expressing, and PVALB-expressing interneurons. 2. Excitatory Neuron Markers: Genes like ATP1A3, CAMK2G, and MAPK1 are often found in excitatory neurons, including cortical layer neurons and pyramidal neurons. 3. Interneuron Markers: SST, VIP, and PVALB are specific markers for different types of interneurons. 4. Glial Cell Markers: Genes such as GFAP, S100B, and AQP4 are typical for astrocytes. Oligodendrocyte markers include MBP, MOG, and PLP1, but none of these are prominently listed. Microglial markers include CX3CR1 and P2RY12, which are not prominently listed. 5. Other Cell Types: Endothelial cells often express genes like PECAM1 and VWF, which are not prominently listed. Given the presence of neuronal markers such as SNAP25, SYT1, and GAD1, the cell is likely a neuron. The presence of GAD1 and GAD2 suggests it is a GABAergic neuron, which points towards an interneuron. Among the interneurons listed, SST-expressing interneurons are a possibility due to the presence of genes like SST and other neuronal markers.
**Final Answer:** SST-expressing interneuron ✗

---

## Drug Response Prediction

**Question:** Below is a list of {gene_names} ordered by descending expression level in a cell. Your task is to predict its response to the drug Erlotinib: [Resistant, Sensitive].
{gene_names}: HSP90AA4P MALSU1 RPL37A FTLP1 PABPC1L RPLP1P13 RPS16P5 RPS14P3......
**Ground Truth:** Sensitive

DeepSeek-R1: The prediction of Erlotinib response hinges on identifying markers linked to resistance or sensitivity. Erlotinib targets EGFR, with sensitivity often tied to activating EGFR mutations (e.g., exon 19 deletions, L858R) and resistance to mechanisms like T790M mutations, MET amplification, or bypass signaling pathways. The provided gene list lacks EGFR itself but includes potential resistance-associated genes: 1. RPS6KA1 (Ribosomal S6 Kinase A1): This kinase is part of the MAPK/ERK pathway downstream of EGFR. High expression suggests sustained ERK signaling independent of EGFR inhibition, a known resistance mechanism. Persistent MAPK pathway activity could bypass EGFR blockade, promoting survival. 2. HSP90 Pseudogenes (HSP90AA4P, HSP90AB2P, HSP90B2P): HSP90 stabilizes oncoproteins, including mutant EGFR. Elevated HSP90 expression might stabilize resistance-associated EGFR variants (e.g., T790M) or other client proteins, enabling survival despite EGFR inhibition. 3. S100A9/S100A11: These genes are implicated in inflammation and resistance pathways, potentially modulating survival signals or drug efflux, though their direct role here is unclear. 4. Ribosomal and Mitochondrial Genes (RPL, RPS, MT-CO3, MT-CYB): High ribosomal activity may indicate proliferation, while mitochondrial genes suggest metabolic adaptation, both common in resistant phenotypes. However, these are broadly expressed and less specific. The absence of clear sensitivity markers (e.g., EGFR mutations, epithelial markers) and the presence of RPS6KA1 (sustaining MAPK signaling) and HSP90 pseudogenes (stabilizing oncoproteins) strongly suggest resistance. While ribosomal and mitochondrial genes are non-specific, their prominence aligns with proliferative and adaptive states typical of resistant cells.
**Final Answer:** Resistant ✗

GPT-4.1: To predict the response to Erlotinib, we need to consider the expression of genes associated with sensitivity or resistance to this drug. Erlotinib is an EGFR (Epidermal Growth Factor Receptor) inhibitor, and sensitivity is often associated with high EGFR pathway activity and low expression of resistance markers such as EMT (epithelial-mesenchymal transition) genes, AXL, MET, or high expression of certain ribosomal and stress response genes. From the provided gene list, the most highly expressed genes are ribosomal proteins (RPL, RPS), heat shock proteins (HSP90AA4P, HSP90AB2P, HSP90B2P), and other housekeeping genes. There is no evidence of high expression of EGFR, AXL, MET, or EMT markers (such as VIM, ZEB1, SNAI1, TWIST1). However, high expression of ribosomal proteins and heat shock proteins can sometimes be associated with a more proliferative, less differentiated state, which may be more sensitive to EGFR inhibition, but high HSP90 expression can also be associated with resistance mechanisms. Notably, there is no clear evidence of canonical resistance markers (e.g., AXL, MET, EMT genes) or sensitivity markers (e.g., high EGFR, ERBB2, or downstream effectors). Given the lack of strong resistance markers and the predominance of ribosomal and housekeeping genes, the cell is more likely to be in a proliferative state, which is generally more sensitive to Erlotinib.
**Final Answer:** Sensitive ✓

GPT-4o: To predict the response of a cell to the drug Erlotinib based on gene expression data, we need to consider the known mechanisms of action of Erlotinib and the role of the genes listed. Erlotinib is an EGFR (epidermal growth factor receptor) inhibitor, and its effectiveness can be influenced by the expression of genes involved in the EGFR signaling pathway, as well as genes related to cell proliferation, apoptosis, and drug resistance mechanisms. In the provided list, there are several ribosomal protein genes (e.g., RPL, RPS), heat shock proteins (e.g., HSP90), and other genes involved in cellular stress responses and protein synthesis. High expression of ribosomal proteins and heat shock proteins can sometimes be associated with increased cellular proliferation and survival, which might contribute to drug resistance. However, without specific information on the expression of genes directly involved in the EGFR pathway or known resistance markers, it is challenging to make a definitive prediction. Generally, high expression of genes that promote cell survival and proliferation could suggest a resistant phenotype, but this is a complex decision that would ideally be informed by more specific data. Given the high expression of heat shock proteins and ribosomal proteins, which can be associated with increased cellular resilience and potential drug resistance, I would lean towards predicting resistance.
**Final Answer:** Resistant ✗

**Question:** Does perturbing NEDD8 in K562 cells cause significant changes in the expression of GFI1B?
**Ground Truth:** Yes

`DeepSeek-R1`: To determine if perturbing NEDD8 in K562 cells significantly alters GFI1B expression, we consider the role of NEDD8 in neddylation, a process critical for cullin-RING ubiquitin ligase (CRL) activity. CRLs mediate protein degradation, and their inhibition via NEDD8 perturbation can stabilize or destabilize transcription factors regulating GFI1B. Experimental evidence, such as studies using the neddylation inhibitor MLN4924 in K562 cells, has shown that GFI1B expression is downregulated following NEDD8 inhibition. This suggests that NEDD8 perturbation disrupts CRL-mediated degradation pathways, indirectly affecting GFI1B levels.
**Final Answer:** Yes ✓

`GPT-4.1`: No.
**Final Answer:** No ✗

`GPT-4o`: Yes.
**Final Answer:** Yes ✓

**Question:** Does perturbing QARS in K562 cells cause the expression of PCM1 to upregulate or downregulate?
**Ground Truth:** Up

`DeepSeek-R1`: To determine whether perturbing QARS in K562 cells affects PCM1 expression, we consider the roles of these genes and potential pathways. QARS, a glutaminyl-tRNA synthetase, is critical for protein synthesis, and its perturbation could activate stress responses like the integrated stress response (ISR) via GCN2 kinase. This might suppress general translation but upregulate stress-related genes. PCM1, involved in centriole assembly, could be influenced by cell cycle changes or structural stress. While direct evidence is lacking, amino acid deprivation stress (via QARS inhibition) often downregulates proliferation-related genes. PCM1's role in cell division suggests potential downregulation. However, stress responses might also upregulate structural genes as compensatory mechanisms. Considering studies on tRNA synthetase inhibition leading to stress-induced gene upregulation, PCM1 might be upregulated. After evaluating these pathways and potential compensatory mechanisms, the balance leans toward upregulation under stress.
**Final Answer:** Up ✓

`GPT-4.1`: Down.
**Final Answer:** Down ✗

`GPT-4o`: Down.
**Final Answer:** Down ✗

# E   Limitation and Future Work

Although CELLVERSE is the first language-centric benchmark for single-cell analysis with LLMs and marks a step forward in applying LLMs to cell biology, it still presents several noteworthy limitations.

First, while CELLVERSE spans hierarchical single-cell analysis tasks, such as cell type annotation (cell-level), drug response prediction (drug-level), and perturbation analysis (gene-level), it does not yet provide a quantitative distinction in their levels of difficulty. Future work could draw inspiration from benchmarks in mathematics [22] to assign difficulty levels to distinct problems, enabling a more nuanced understanding of the problem-solving capabilities of LLMs in cell biology.

Second, although CELLVERSE introduces a pipeline for converting raw single-cell multi-omics data into natural language QA problems across multiple tasks, all prompts and questions are currently formulated in English. Extending the benchmark to include multilingual QA settings would improve its global applicability and allow for more comprehensive evaluation of LLMs in terms of linguistic diversity and understanding, which we leave as future work.

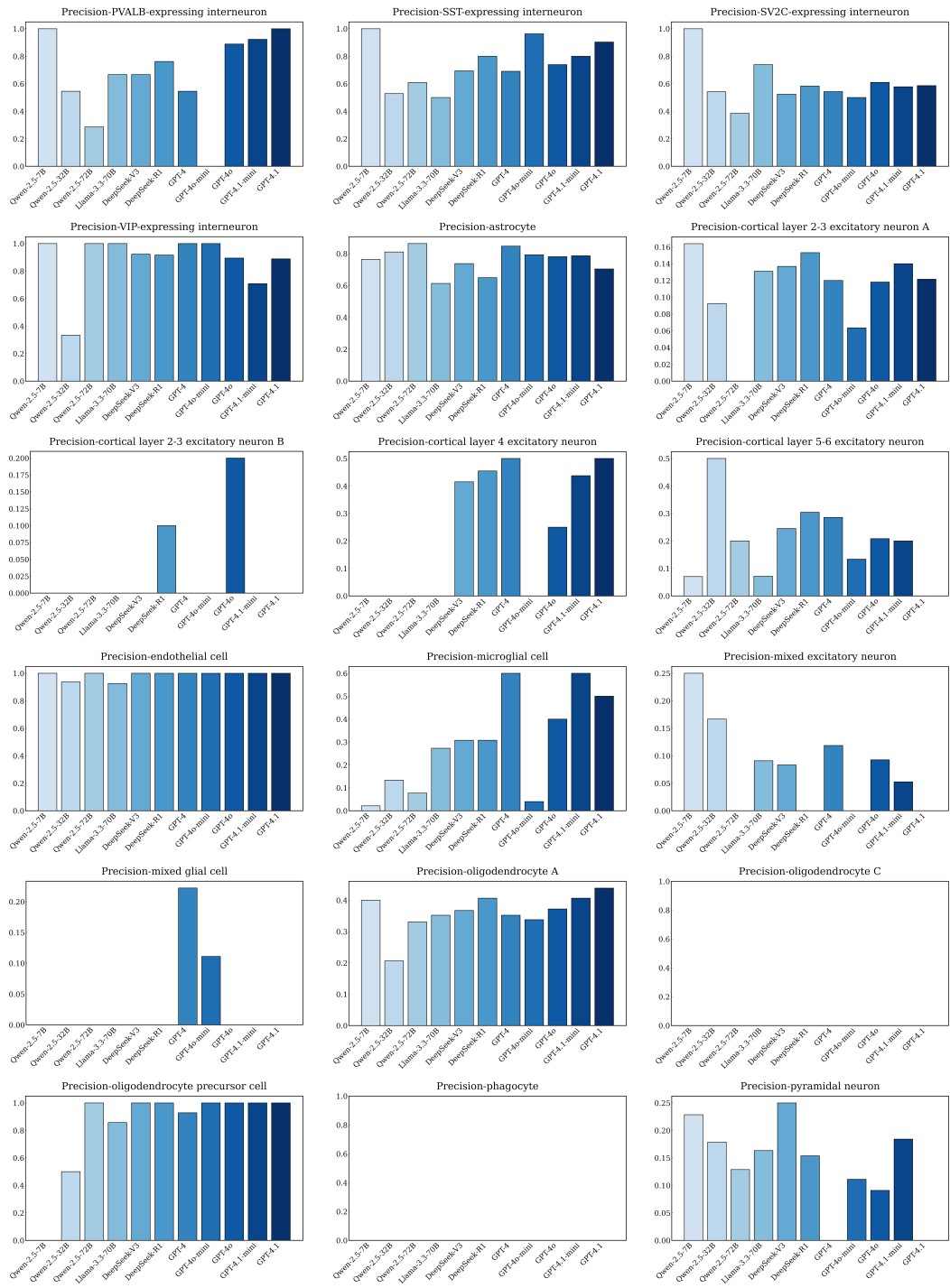

Figure 8: Precision score comparison of specific cell types on scRNA-seq data. All LLMs fail to predict certain challenging cell types, such as oligodendrocyte C and phagocyte.

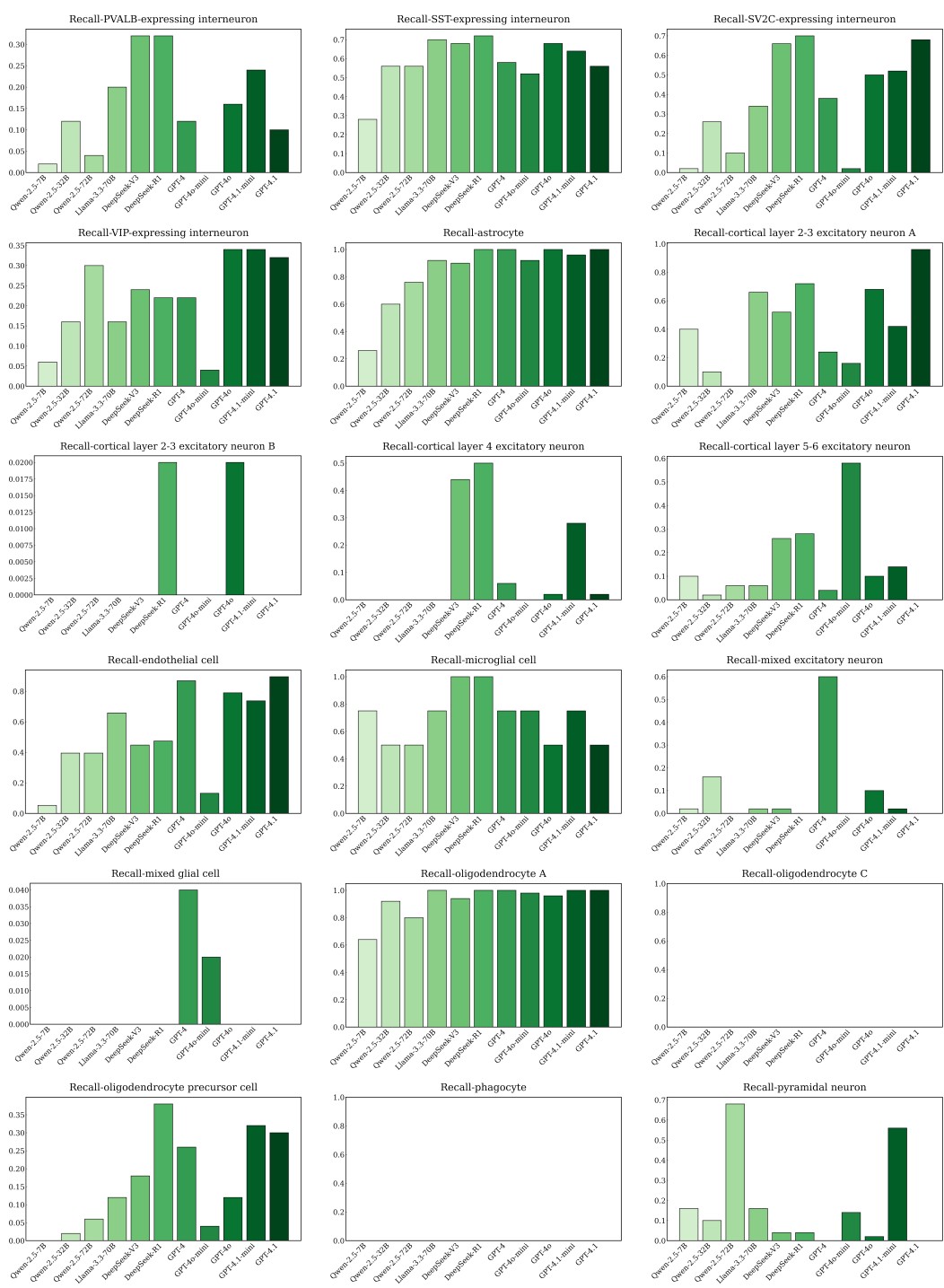

Figure 9: Recall score comparison of specific cell types on scRNA-seq data. All LLMs fail to predict certain challenging cell types, such as oligodendrocyte C and phagocyte.

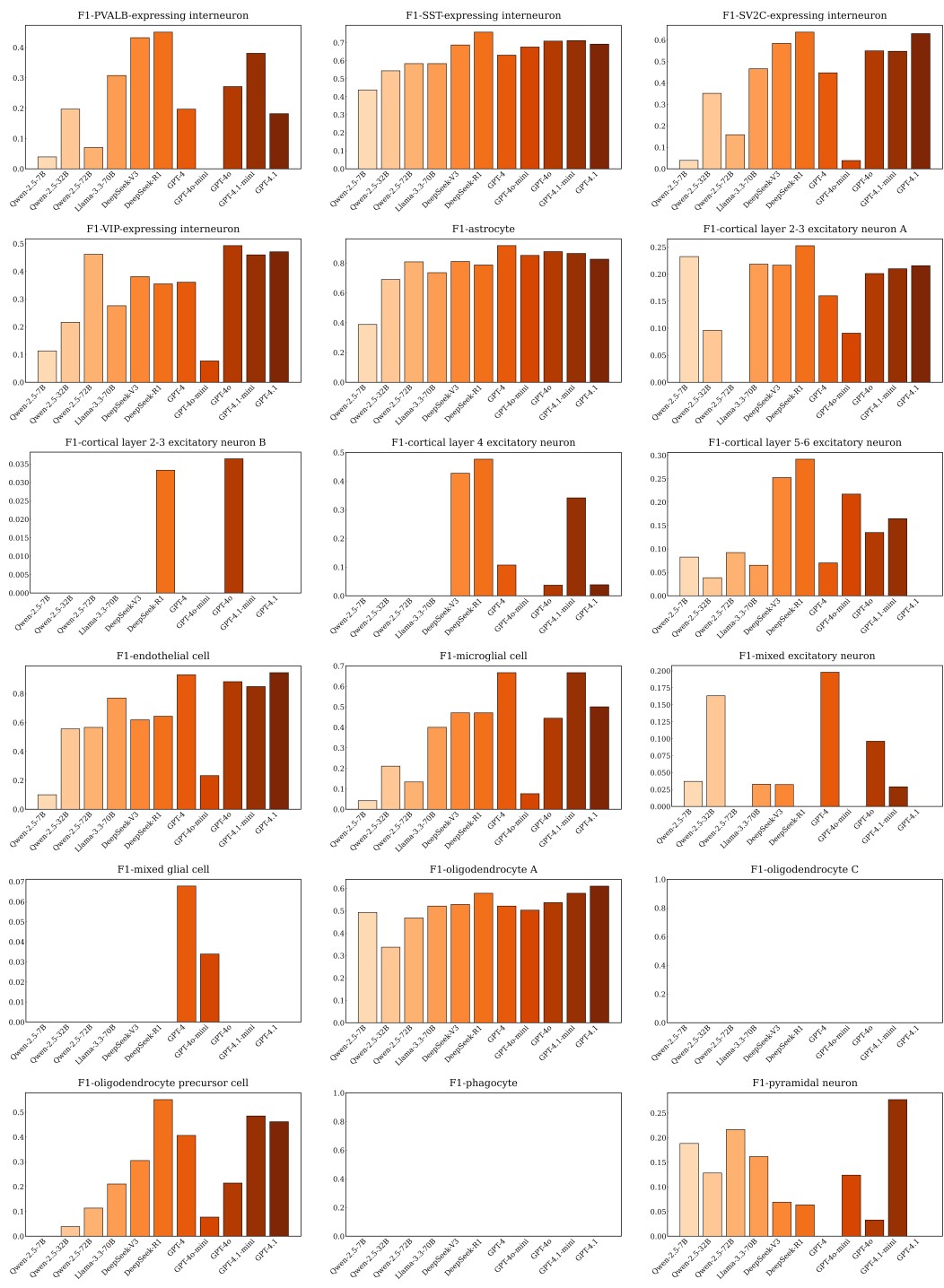

Figure 10: F1 score comparison of specific cell types on scRNA-seq data. All LLMs fail to predict certain challenging cell types, such as oligodendrocyte C and phagocyte.

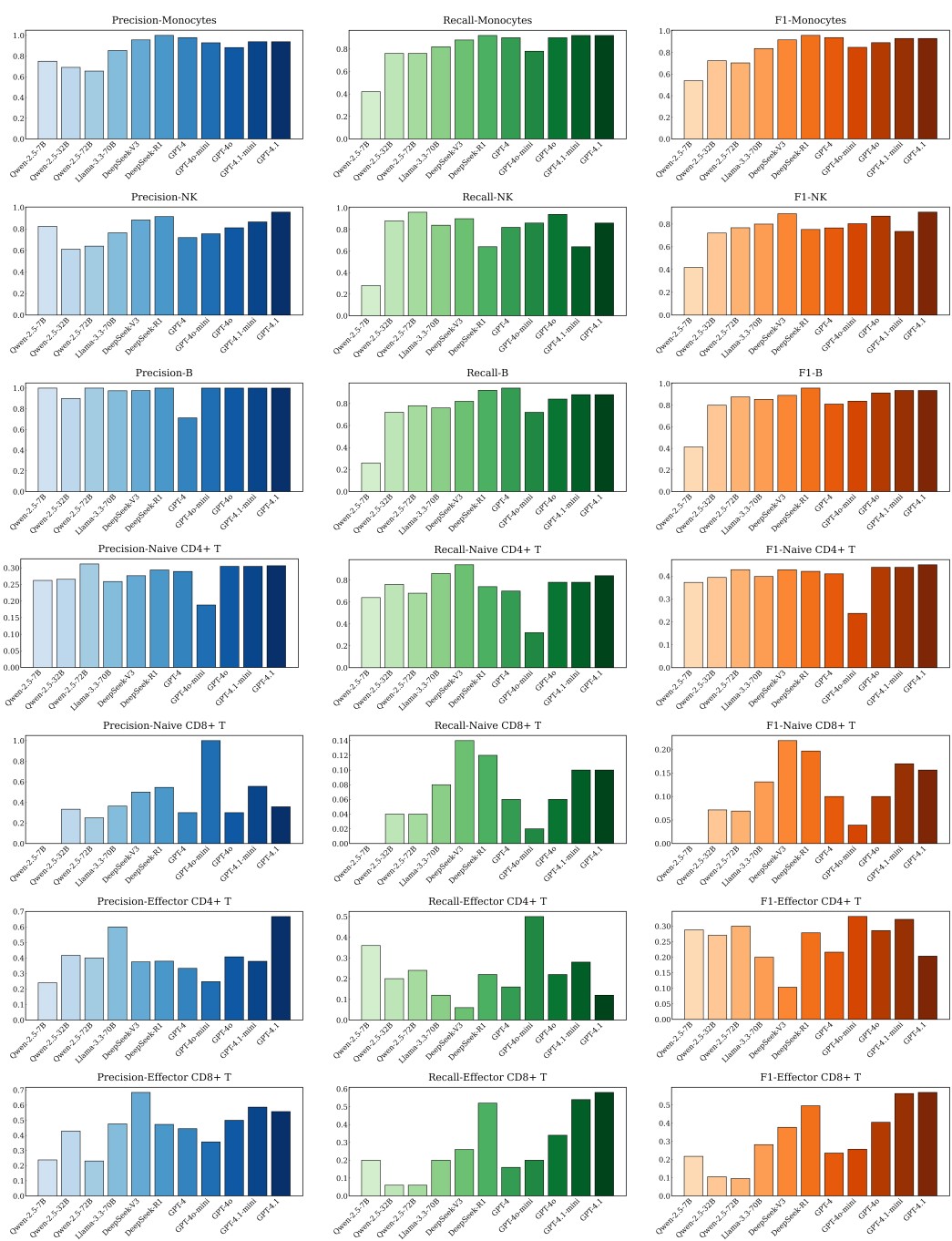

Figure 11: Performance comparison of specific cell types on CITE-seq data.

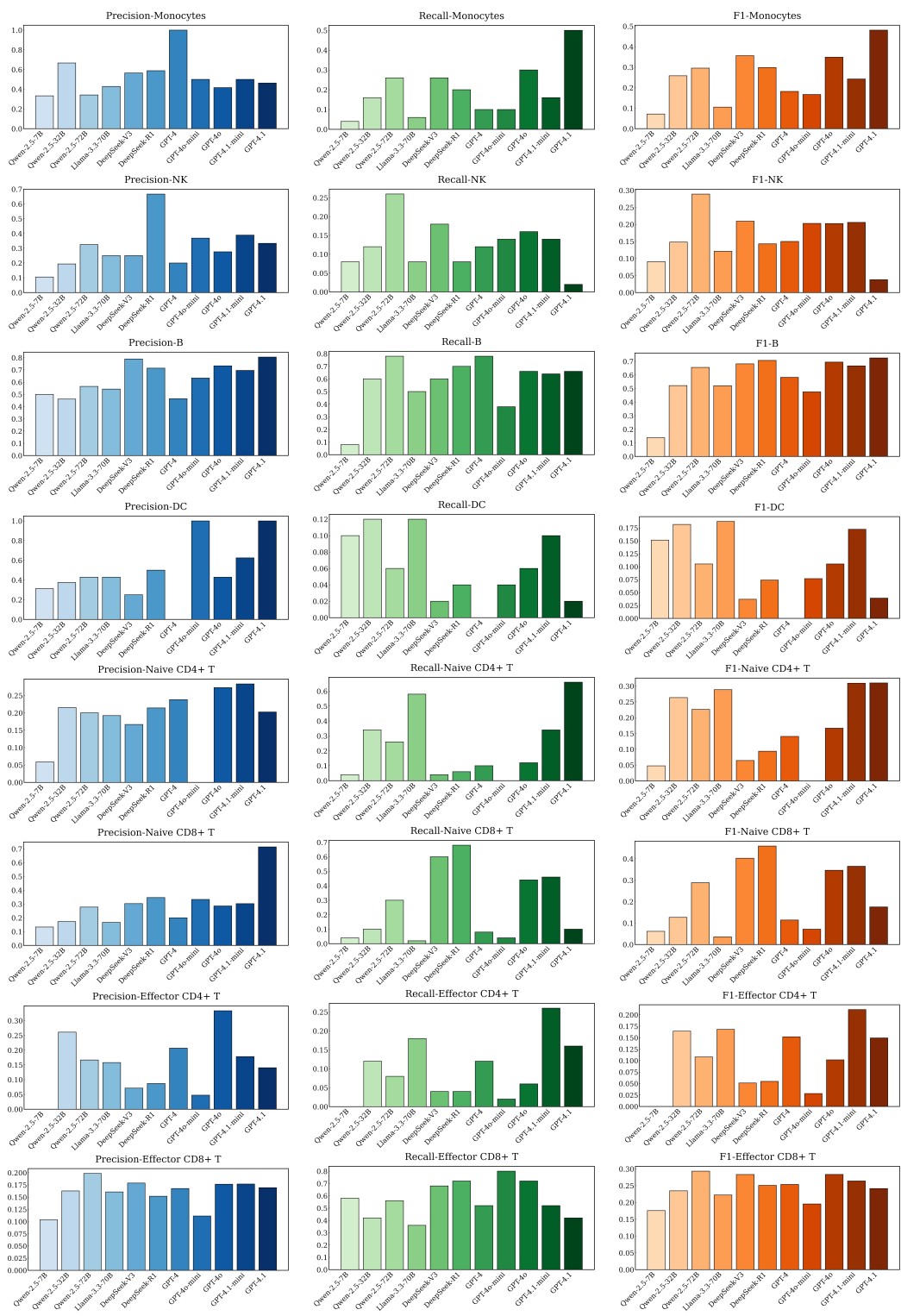

Figure 12: Performance comparison of specific cell types on ASAP-seq data.

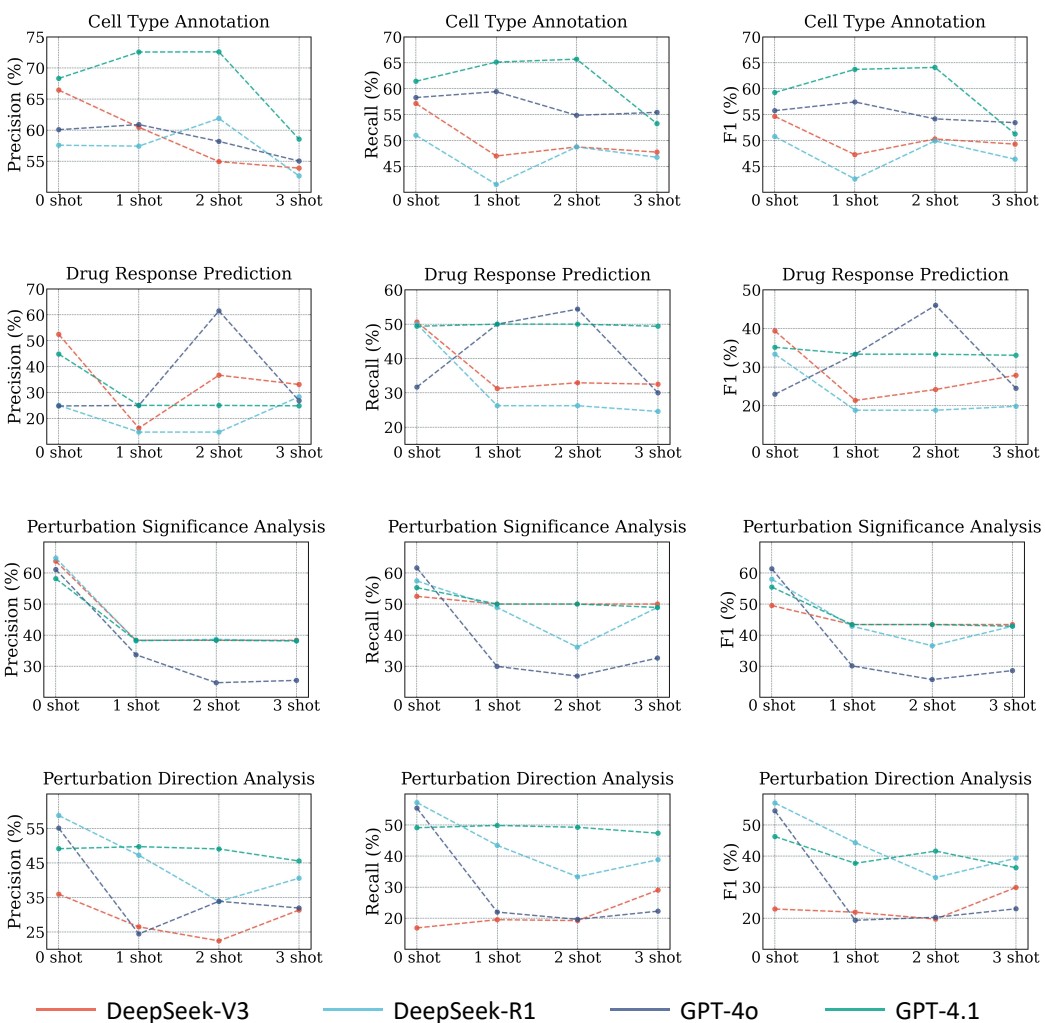

Figure 13: Precision, Recall, and F1 score comparison under few-shot settings.

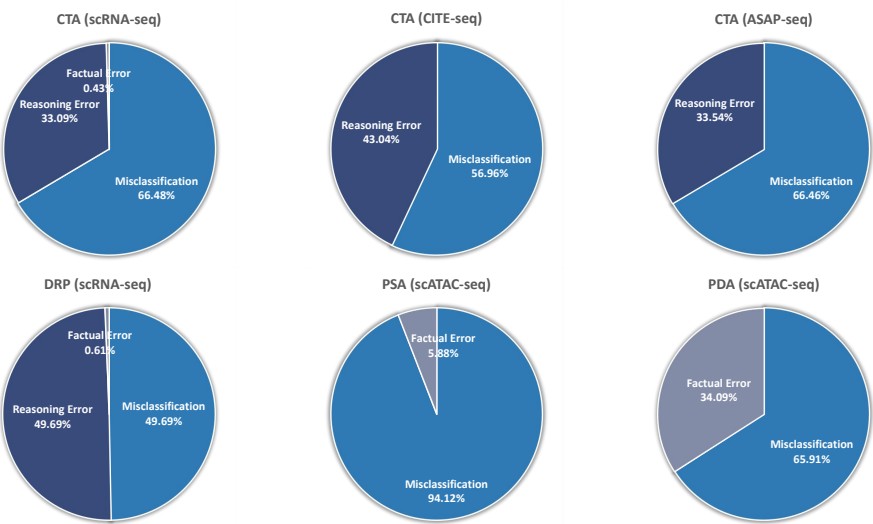

Figure 14: Distribution of `GPT-4.1`'s errors within distinct types across various tasks.

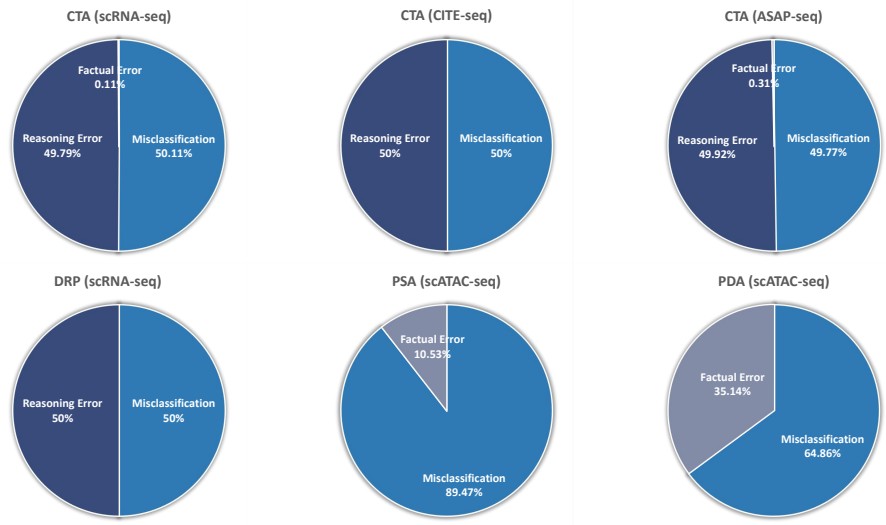

Figure 15: Distribution of `GPT-4o`'s errors within distinct types across various tasks.

