# OpenReview forum: "CellVerse: Do Large Language Models Really Understand Cell Biology?"
_NeurIPS.cc/2025/Datasets_and_Benchmarks_Track — NeurIPS 2025 Datasets and Benchmarks Track poster_

### Official Review · Reviewer_A5e3 · 2025-06-04

**Rating:** 6
**Confidence:** 5

**Summary:**

This paper proposes a novel language-driven paradigm for single-cell analysis and introduces CellVerse, the first dataset of its kind in this domain. CellVerse is a unified, user-friendly, and interpretable single-cell analysis dataset that encompassing a variety of single-cell omics data (scRNA-seq, CITE-seq, ASAP-seq, and scATAC-seq data) and a suite of hierarchical tasks (cell type annotation, drug response prediction, and perturbation analysis). Through extensive evaluation across 14 state-of-the-art LLMs, it reveals both the capabilities and limitations of current LLMs in understanding cell biology.

**Additional Feedback:**

Please refer to the limitation section.

**Dataset Code Accessibility:**

Yes

**Dataset Code Comments:**

I have taken a closed look at the github repo, project page, and huggingface dataset provided by the authors. I think the dataset and code are avalable in a usable format and well documented. The proposed benchmark dataset is fully open-sourced and will be easily followed.

**Ethical Comments:**

This paper introduces a novel paradigm for single-cell analysis and offers the first large-scale empirical validation. This benchmark can facilitie the area of LLMs for cell biology and perhaps inspire some research progress in the future. I have taken a closed look at the provided data and code, and find no ethical concerns.

**Ethical Considerations:**

No, there are no or only very minor ethics concerns

**Final Justification:**

The author’s thorough response and the new experimental results have resolved all of my previous concerns.

**Limitations Weaknesses:**

I have not see any obvious weaknesses of this research. However, I have some additional questions and suggestions for improving this paper：
1. This paper lacks a discussion about the computational cost of the evaluated LLMs, especially for open-sourced LLMs. Incorporating this discussion could offer more insights for subsequent researchers and readers.
2. What was the rationale behind selecting the specific LLMs for evaluation?
3. Why did C2S-Pythia fail across all tasks, and what are the potential areas for improvement?
4. Did the prompt strategies vary for different LLMs?

**Strengths Contributions:**

1. The paper is generally well-written and well-structured. The figures are visually appealing, making the content easy to read and understand.
2. This paper offers a new perspective and a benchmarking foundation for research in single-cell analysis, paving the way for a new paradigm in cell biology. The proposed cell-to-text transformation using C2S and GRN is a direct and reliable approach.
3. The experiments are sufficiently comprehensive. The proposed dataset includes a wide range of tasks spanning the cell, drug, and gene levels, and integrates multi-omics single-cell data, laying a solid foundation for future research.
4. The analysis is thorough, including both few-shot and scaling context length evaluations. It also outlines promising directions for future work.

---

> ### Author Rebuttal · Authors · 2025-07-30
>
> We sincerely appreciate the time you've taken to review our paper and for your insightful comments. Your positive feedback is highly encouraging for us! We'd like to address your concerns in the following response.
>
> > Q1. This paper lacks a discussion about the computational cost of the evaluated LLMs, especially for open-sourced LLMs. Incorporating this discussion could offer more insights for subsequent researchers and readers.
>
> A1. We sincerely appreciate the reviewer's valuable suggestion regarding computational cost analysis. This is indeed an important consideration for practical implementation. To provide clarity on hardware requirements for running these open-sourced models:
> - For models <8B parameters (like LLaMA-3.1-8B), single-GPU inference (bfloat16 precision) is feasible with a consumer-grade 40GB GPU (e.g., A6000).
> - For ∼30B models (like Qwen-2.5-32B), single-GPU inference is achievable using high-end 80GB GPUs (e.g., A100/H100).
> - For ∼70B models (like LLaMA-3.3-70B), dual-GPU inference requires 2×80GB devices with tensor parallelism.
>
> Thank you for highlighting this important dimension and we will also incorporate a dedicated subsection detailing these computational considerations in our revised manuscript.
>
> > Q2. What was the rationale behind selecting the specific LLMs for evaluation?
>
> A2. Thank you for the thoughtful question. The selection of LLMs for evaluation was based on the **state-of-the-art** models available **at the time of completing the paper**, including both **leading closed-source and open-source** LLMs. We also included the only available specialist model family (the C2S series) to provide a comparison between general-purpose and biology-focused models.
>
> We will clarify this rationale in the revised version. Thank you again for your feedback.
>
> > Q3. Why did C2S-Pythia fail across all tasks, and what are the potential areas for improvement?
>
> A3. Thank you for raising this important question. The underperformance of C2S-Pythia across all tasks can be attributed to several key factors:
>
> - **Model scale limitations**
> C2S-Pythia has a relatively small parameter size compared to modern state-of-the-art LLMs. Using larger models (e.g., 7B parameters or more) may significantly improve performance.
> - **Domain-restricted training data**
> The model was trained exclusively on biological domain data, which limits its general language understanding and reasoning capabilities. Incorporating general-domain corpora or continuing training on top of a strong general-purpose LLM could enhance both comprehension and generalization capability.
> - **Lack of post-training alignment techniques**
> Techniques like reinforcement learning from human feedback (RLHF) have proven effective in improving LLM generalization and usability. Integrating such post-training alignment techniques could help C2S-Pythia better capture task intent and improve response quality.
>
> We will expand on these points in the revised version to provide a clearer analysis of C2S-Pythia’s limitations and potential future directions.
>
> > Q4. Did the prompt strategies vary for different LLMs?
>
> A4. Thank you for your question. To ensure **fairness and consistency**, we used the same prompt strategies across all LLMs evaluated in our benchmark. This allows for a more controlled and comparable assessment of model performance, without introducing confounding factors related to prompt design.
>
> We will clarify this point in the revised version. Thank you again for bringing it up.
>
> Thanks again for appreciating our work and for your constructive suggestions. Please let us know if you have further questions.

---

> > ### Comment · Reviewer_A5e3 · 2025-08-01
> >
> > Thanks for your detailed reply and the additional experimental results, which have resolved all of my earlier concerns.

---

> > > ### Author Response · Authors · 2025-08-04
> > > **Thanks for your effort and feedback!**
> > >
> > > Thanks once more for your time and feedback on our work. We are delighted that the additional information/results resolved your concerns. Rest assured, we will integrate all rebuttal content into the revised manuscript according to your helpful suggestions.

---

### Official Review · Reviewer_tCEs · 2025-06-27

**Rating:** 5
**Confidence:** 5

**Summary:**

This paper introduces CELLVERSE, a novel, unified language-centric benchmark for evaluating Large Language Models (LLMs) on single-cell biology analysis tasks. The authors address key limitations in existing single-cell analysis paradigms (lack of unification, poor user-friendliness, limited interpretability) by reformulating single-cell data (scRNA-seq, CITE-seq, ASAP-seq, scATAC-seq) into natural language formats using Cell2Sentence (C2S) and Gene Regulatory Networks (GRNs). CELLVERSE encompasses three hierarchical tasks: cell type annotation (cell-level), drug response prediction (drug-level), and perturbation analysis (gene-level). The authors conduct a comprehensive evaluation of 14 open-source and closed-source LLMs (ranging from 160M to 671B parameters), including specialist models (C2S-Pythia) and generalist models (Qwen, Llama, GPT, DeepSeek families). Key findings reveal that specialist models fail across tasks due to hallucinations and limited capacity, while generalist models show preliminary reasoning capabilities but fall significantly short of expectations. CELLVERSE provides the first large-scale empirical evidence of the significant challenges in applying LLMs to cell biology and establishes a foundation for future research.

**Additional Feedback:**

NA

**Dataset Code Accessibility:**

Yes

**Dataset Code Comments:**

The data and code are accessible and well-documented for practical use. This open-source benchmark enables straightforward implementation.

**Ethical Comments:**

Exploring the ability of LLMs to comprehend cell biology is highly compelling and could draw in more researchers to this field, advancing the LLMs for Science community. To date, I have not identified significant ethical concerns or negative societal impacts.

**Ethical Considerations:**

No, there are no or only very minor ethics concerns

**Final Justification:**

Dear AC, I will keep my rating as my initial score 5: Accept. I do believe this submission would be a good addition to the research community. The rebuttal firms my recommendation.

**Limitations Weaknesses:**

1. Emphasizing the "no better than random" result for drug response prediction in the abstract/conclusion even more strongly is warranted, as it's a particularly striking and critical finding.

2. Consider adding horizontal lines indicating "random baseline" performance (e.g., 50% for binary tasks, 1/N for N-class tasks) to the leaderboard figure (Fig 4) to make the comparison against chance even more visually immediate.

3. The future work points are excellent. Explicitly adding "investigating hybrid approaches (LLMs + traditional ML/domain models)" or "incorporating structured biological knowledge bases more deeply" could be valuable extensions.

**Strengths Contributions:**

1. CELLVERSE is a major contribution as the first comprehensive, language-centric benchmark specifically designed to evaluate LLMs on core single-cell analysis tasks. It fills a critical gap in the field.

2. The transformation of diverse single-cell multi-omics data (4 types) into a unified QA format via C2S and GRNs is innovative and directly addresses the identified limitations of current approaches (unification, user-friendliness, interpretability). This paradigm shift holds significant potential.

3. The scale and scope of the evaluation are impressive. Assessing 14 diverse LLMs (including cutting-edge closed-source models like GPT-4.1) across 4 data types and 3 task levels provides a robust and valuable snapshot of current capabilities.

4. The paper is well-structured, clearly written, and effectively uses figures/tables to summarize the benchmark design, data statistics, and extensive results.

---

> ### Author Rebuttal · Authors · 2025-07-30
>
> We sincerely appreciate the time you've taken to review our paper and for your insightful comments. Your positive feedback is highly encouraging for us! We'd like to address your concerns in the following response.
>
> > W1. Emphasizing the "no better than random" result for drug response prediction in the abstract/conclusion even more strongly is warranted, as it's a particularly striking and critical finding.
>
> A1. Thank you for the helpful suggestion. We agree that the near-random performance on drug response prediction is a particularly striking result that highlights the current limitations of LLMs in complex biological tasks. We will emphasize this finding more clearly in the abstract and conclusion of the revised manuscript.
>
>
> > W2. Consider adding horizontal lines indicating "random baseline" performance (e.g., 50% for binary tasks, 1/N for N-class tasks) to the leaderboard figure (Fig 4) to make the comparison against chance even more visually immediate.
>
> A2. Thank you very much for your helpful suggestion. We have added the random baseline performance with results shown in the table below. We will alos add this into the revised version.
>
> | Task| CTA (scRNA-seq) | CTA (CITE-seq) | CTA (ASAP-seq) | DRP | PSA | PDA |
> | -| - | - | -|- | - | -|
> | **Precision**| 5.08 | 15.19 | 11.53|49.37 | 51.01 | 44.92 |
> | **Recall**| 4.29 | 14.86 | 11.56|49.37 | 51.40 | 44.56 |
> | **F1**| 4.61 | 14.96 | 11.51|49.36 | 45.79 | 43.75 |
> | **Acc**| 5.08 | 14.86 | 11.56|49.38 | 48.33 | 44.44 |
>
> > W3. The future work points are excellent. Explicitly adding "investigating hybrid approaches (LLMs + traditional ML/domain models)" or "incorporating structured biological knowledge bases more deeply" could be valuable extensions.
>
> A3. Thank you for the thoughtful suggestion. We agree that hybrid approaches combining LLMs with traditional machine learning or domain-specific models, as well as deeper integration of structured biological knowledge bases, are highly promising directions. We will add these points explicitly to the future work section in the revised version.
>
> Thanks again for appreciating our work and for your constructive suggestions. Please let us know if you have further questions.

---

> > ### Comment · Reviewer_tCEs · 2025-08-01
> >
> > Thanks to the authors for their efforts during the rebuttal period. All of my questions have been solved. In addition, I have carefully reviewed the authors' responses to other reviewers’ comments. I think this is a great benchmark work that can inspire the paradigm of next-generation single-cell analysis.

---

> > > ### Author Response · Authors · 2025-08-04
> > > **Thanks for appreciating our work!**
> > >
> > > Thanks for your positive feedback and for appreciating our work! We are pleased to know that your concerns have been addressed. We will properly include all the rebuttal contents in the revised version, following your valuable suggestions.

---

### Official Review · Reviewer_Pgyi · 2025-07-03

**Rating:** 5
**Confidence:** 2

**Summary:**

This paper introduces CELLVERSE, a unified, language-centric benchmark designed to evaluate the reasoning capabilities of large language models (LLMs) in single-cell biology. By transforming single-cell multi-omics data (scRNA-seq, scATAC-seq, CITE-seq, ASAP-seq) into question-answering (QA) problems, CELLVERSE allows systematic evaluation of LLMs on three hierarchical tasks: cell type annotation, drug response prediction, and perturbation analysis. The authors test 14 LLMs (both open- and closed-source, ranging from 160M to 671B parameters) and find that while generalist models show emerging capabilities, their overall performance remains far from satisfactory. CELLVERSE is positioned as the first large-scale benchmark connecting LLMs to single-cell data analysis, with the intention to lay a foundation for advancing natural language-driven cell biology research.

**Dataset Code Accessibility:**

Yes

**Ethical Considerations:**

No, there are no or only very minor ethics concerns

**Final Justification:**

Consider the rebuttal, my concern regarding the performance and task are resolved. I will consider this paper as accept (5).

**Limitations Weaknesses:**

- **Performance Still Low**: Although the benchmark is promising, the results (e.g., accuracies barely above random guessing for drug response prediction in Table 2) indicate that current LLMs perform poorly. The paper could analyze *why* LLMs fail on these tasks in more depth (e.g., feature redundancy, data sparsity, missing biological priors).
- **Task Difficulty Calibration**: The benchmark treats all questions as equally challenging. Adding difficulty levels or “challenge tiers” would help researchers diagnose model weaknesses more precisely.
- **Limited Multilingual Consideration**: The benchmark is entirely English-only, though the authors do acknowledge this in future work (Appendix E). Supporting multilingual questions would improve global applicability.
- **Closed-Source Model Dependency**: Many leaderboard results rely on proprietary models (GPT-4 variants, etc.), which somewhat limits reproducibility and community-driven progress. Strengthening open-source baselines would be valuable.
- **Limited Error Taxonomy**: The error analysis (Figures 7, 14, 15) could go deeper, for instance highlighting errors on rare cell types or genes of high clinical relevance, to improve interpretability for biologists.

Overall, these are fixable limitations, and many are already acknowledged by the authors.

**Strengths Contributions:**

- **Novel Benchmark**: The introduction of CELLVERSE fills a clear gap in evaluating LLMs for data-intensive, domain-specific biological tasks, especially single-cell multi-omics analysis. The idea of translating high-dimensional omics data into natural-language QA tasks is compelling and novel (Sec. 1, Sec. 3).
- **Broad Coverage**: CELLVERSE supports multiple omics data types and three representative single-cell analysis tasks across hierarchical levels (cell-, drug-, gene-level), giving it strong breadth and relevance (Sec. 3.1, Fig. 2).
- **Systematic Evaluation**: The experimental evaluation is thorough, covering 14 strong LLMs under consistent conditions (Sec. 4), providing a valuable empirical baseline. Tables 1–4 and Figures 4–7 are informative, clear, and well-labeled.
- **Transparent Reporting**: The paper includes extensive details on dataset curation, prompting strategies, evaluation protocols, and even error analysis (Appendix sections), promoting reproducibility and transparency.
- **Motivated by Real Bottlenecks**: The paper convincingly articulates why existing single-cell pipelines suffer from lack of unification, poor interpretability, and steep domain-expertise barriers (Sec. 1, Fig. 1), and why a language-based interface could improve accessibility.
- **Potential Impact**: CELLVERSE provides a rigorous testbed that can stimulate future research at the intersection of language models and biological reasoning, with positive implications for making single-cell analysis more interpretable and user-friendly.

Overall, the paper is well-written, clearly organized, and meets the expectations for a high-quality Datasets & Benchmarks submission.

---

> ### Author Rebuttal · Authors · 2025-07-30
>
> We greatly appreciate your time in reviewing our paper and your insightful comments. Your positive feedback is incredibly encouraging for us! We'd like to address your concerns in the following response.
>
> > W1. **Performance Still Low:** Although the benchmark is promising, the results (e.g., accuracies barely above random guessing for drug response prediction in Table 2) indicate that current LLMs perform poorly. The paper could analyze why LLMs fail on these tasks in more depth (e.g., feature redundancy, data sparsity, missing biological priors).
>
> A1. We sincerely thank the reviewer for pointing out this important aspect of our work. This observation is indeed central to our motivation: our benchmark is designed not only to evaluate existing models, but also to highlight current limitations and challenges in applying LLMs to cell biology.
>
> We believe that the low performance stems from multiple underlying factors, among which the most significant is the inherent nature of single-cell data:
>
> - **High technical noise:** Single-cell omics data is notoriously noisy due to stochastic gene expression and measurement variability.
> - **Strong batch effects:** Differences across datasets or experimental conditions introduce systematic biases, which hinder model generalization.
>
> These characteristics make cell biology a particularly challenging domain for LLMs, and underscore the need for further innovation in model design, domain adaptation, and integration of biological structure or priors.
>
> We appreciate the reviewer’s suggestion and will include a more in-depth analysis of these failure cases and underlying challenges in the revised version. We believe this will further clarify the diagnostic value of our benchmark and motivate future work in this emerging intersection of LLMs and biological data. Thank you again for your thoughtful and constructive feedback.
>
>
> > W2. **Task Difficulty Calibration:** The benchmark treats all questions as equally challenging. Adding difficulty levels or “challenge tiers” would help researchers diagnose model weaknesses more precisely.
>
> A2. Thank you very much for your thoughtful suggestion regarding task difficulty calibration. We agree that this is a valuable direction, and we appreciate you highlighting it. In future work, we will explore ways to assign difficulty labels to each question, potentially based on:
> - **Annotation heuristics** (e.g., answer length, number of reasoning steps required).
> - **Empirical difficulty** (e.g., based on human or model performance distributions).
> - **Biological complexity** (e.g., involving rare cell types or subtle gene-drug interactions).
>
> We will note this as an important direction in our revised version, and aim to include such difficulty annotations in future iterations of the benchmark. Thank you again for your constructive and insightful feedback.
>
> > W3. **Limited Multilingual Consideration:** The benchmark is entirely English-only, though the authors do acknowledge this in future work (Appendix E). Supporting multilingual questions would improve global applicability.
>
> A3. Thank you for your thoughtful comment on the language scope of our benchmark. We acknowledge that the current version is limited to English, which is primarily due to the fact that most mainstream genomics and cell biology databases and ontologies are curated in English. As a result, the source data and domain-specific terminology naturally bias the benchmark toward an English-only setting.
>
> That said, we fully agree that supporting multilingual queries would enhance the global accessibility and inclusiveness of the benchmark. We will explore this direction in future work, including **possible extensions to other major languages** and **cross-lingual evaluation settings**.
>
> Thank you again for highlighting this important point.
>
> > W4. **Closed-Source Model Dependency:** Many leaderboard results rely on proprietary models (GPT-4 variants, etc.), which somewhat limits reproducibility and community-driven progress. Strengthening open-source baselines would be valuable.
>
> A4. Thanks for your helpful suggestion. We have included three more cutting-edge open-source LLMs for performance comparison, with results shown in the table below. We will also add this in the revised manuscript.
>
> | scRNA-seq | Precision | Recall |F1 |Acc  |
> | --|--|--|--|--|
> | LLaMA-3.1-8B| 21.54 | 11.82   |8.16   |11.90 |
> | Qwen-3-8B   | 32.15 | 24.90   |19.57  |25.27 |
> | Qwen-3-14B  | 41.41 | 28.72   |27.59  |32.22 |
>
> | CITE-seq | Precision | Recall |F1 |Acc  |
> | --|--|--|--|--|
> | LLaMA-3.1-8B| 35.49 | 20.50   |22.02  |23.43 |
> | Qwen-3-8B   | 44.68 | 39.50   |39.98  |45.14 |
> | Qwen-3-14B  | 55.44 | 52.29   |50.40  |52.29 |
>
> | ASAP-seq | Precision | Recall |F1 |Acc  |
> | --|--|--|--|--|
> | LLaMA-3.1-8B| 9.70 | 2.60   |3.93   |2.89 |
> | Qwen-3-8B   | 24.70 | 18.20  |16.06 |20.22 |
> | Qwen-3-14B  | 25.77 | 21.00  |19.85  |23.33 |
>
>
> > W5. **Limited Error Taxonomy:** The error analysis (Figures 7, 14, 15) could go deeper, for instance highlighting errors on rare cell types or genes of high clinical relevance, to improve interpretability for biologists.
>
> A5. Thank you for the insightful suggestion. We have included a discussion on in-depth error analysis about specific rare cell types:
>
> ```
> Question:
> Below is a list of {gene_names} ordered by descending expression level in a single-cell CITE-seq sample. Your task is to select one cell type to which this cell most likely belongs from the following candidate cell types: {cell_types}. Make your choice in format 'Final Answer: Prediction'.
> {gene_names}: MALAT1 TMSB4X TPT1 RPL13 RPL10 RPS19 ACTB RPS12 RPS14 IL32 RPS3A LTB RPL30 RPLP1 RPL28 RPL13A RPS27 RPS3 RPS7 RPL3 HLA-B RPL14 RPS13 UBA52 RPS6 RPS23 RPS9 RPL11 HLA-C RPS15A RPS24 RPL6 RPL36 RPL37A HLA-A RPS21 RPS15 S100A4 RPL39 CYBA RPL29 RPL18A S100A6 PTPRC RPL22 PTMA NACA HLA-E RPS20 RPS29 RPL27 RPL9 EIF1 SERF2 CFL1 RPL38 HSP90AB1 RPL23A STAT1 RPL37 IFITM2 CALM1 ATP5F1E KIAA1551 ISG15 SON HLA-F PABPC1 CD48 YWHAZ PPIA RPL10A RPL5 HNRNPA2B1 STK17A HINT1 CALR UBB FTH1 RPSA GAPDH CORO1A PSME1 GIMAP7 XAF1 ZNF217 FXYD5 CD2 GADD45GIP1 NCL CYSLTR1 RPL24 RPL31 SAMD9 TRGC2 ENO1 TAF15 CDC42 PSMA7 SSB
> {cell_types}: 'Naive CD4+ T', 'Effector CD4+ T', 'Naive CD8+ T', 'NK', 'Effector CD8+ T', 'B', 'Monocytes'.
>
> DeepSeek-R1 Response:
> The gene list provided includes markers indicative of T cell identity (e.g., IL32, LTB, PTPRC) and lacks classical markers for B cells (e.g., CD79A, MS4A1), monocytes (e.g., CD14, LYZ), or NK cells (e.g., NKG7, GNLY). Among T cell subsets, IL32 is strongly associated with activated/effector CD4+ T cells, while the absence of cytotoxic genes (e.g., GZMB, PRF1) and CD8-specific markers (e.g., CD8A/B) makes CD8+ effector T cells less likely. Additionally, S100A4 and STAT1 suggest an activated/effector phenotype, further supporting effector CD4+ T cells over naive subsets (which typically express CCR7 or SELL).
> Final Answer: Effector CD4+ T
>
> Ground Truth:
> Feector CD8+ T
> ```
>
> #### **Error Analysis:**
> The model incorrectly predicted *"Effector CD4+ T"* instead of the ground truth *"Effector CD8+ T"* due to:
> 1. **Biased interpretation of `IL32`** as CD4+-specific,
> 2. **Inadequate handling of single-cell data limitations** (dropout effects),
> 3. **Failure to recognize CD8+-specific signatures** in the expression profile.
>
> #### **Detailed Breakdown of Errors:**
>
> ##### 1. **Misattribution of `IL32` as CD4+ T Cell-Specific**
> - **Model's flawed claim**:
>   > *"IL32 is strongly associated with activated/effector CD4+ T cells"*
> - **Reality**:
>   - `IL32` is a **general T cell activation marker** expressed in both **CD4+ and CD8+ effector subsets**.
>   - High `IL32` occurs in antiviral CD8+ T cells (e.g., in COVID-19/HIV infection models) and tumor-infiltrating CD8+ T cells.
> - **Impact**: Using `IL32` to exclude CD8+ T cells was biologically invalid.
>
> ##### 2. **Faulty Exclusion of CD8+ T Cells**
> - **Model's flawed reasoning**:
>   > *"Absence of cytotoxic genes (e.g., `GZMB`, `PRF1`) and CD8-specific markers (e.g., `CD8A/B`)..."*
> - **Critical oversights**:
>   - **Technical noise**: Single-cell RNA-seq frequently suffers from dropout effects for low-abundance genes like `CD8A/B`. Absence ≠ biological irrelevance.
>   - **Effector heterogeneity**: Not all CD8+ effectors express cytotoxic genes. *Interferon-responsive subsets* dominate when `ISG15`/`STAT1`/`IFITM2` are upregulated.
>   - **Missed CD8+ evidence**:
>     - `TRGC2` (TCR γ-chain) hints at cytotoxic lineage,
>     - `IFITM2` is CD8+-enriched,
>     - `HLA-A/B/C` (MHC-I molecules) engage CD8+ T cells.
>
> ##### 3. **Ignoring Key CD4+ T Cell Counterevidence**
> - **Model's inconsistency**:
>   - Cited absence of `CD8A/B` to exclude CD8+ cells, but **ignored absence of CD4+ markers**:
>     - No `CD4`, `IL7R` (CD127), or `CCR7` (naive marker) in the gene list.
>   - Overvalued non-specific activation markers (`S100A4`, `STAT1`), which cannot distinguish CD4+ from CD8+ effectors.
>
> ##### 4. **Underweighting CD8+-Supportive Signatures**
> | Gene  | Role | Relevance to CD8+ T Cells |
> |--|--|--|
> | **`ISG15`**| Interferon-stimulated gene | Hallmark of antiviral CD8+ response |
> | **`STAT1`**| IFN signaling transducer | Drives CD8+ effector differentiation|
> | **`IFITM2`**| Viral restriction protein | Enriched in activated CD8+ T cells |
> | **`HLA-*`** | Antigen presentation | Required for CD8+ T cell engagement |
>
> ##### 5. **Misinterpretation of Activation Signatures**
> - **Model confusion**: Interpreted ribosomal genes (`RPL*`/`RPS*`) and `S100A4` as "CD4+-specific activation" evidence.
> - **Reality**: These indicate *general proliferative/activated states* and are **equally abundant in CD8+ effectors**.
>
> We will also added a more detailed and comprehensive version in the revised manuscript.
>
> Thanks again for appreciating our work and for your constructive suggestions. Please let us know if you have further questions.

---

> > ### Comment · Reviewer_Pgyi · 2025-08-08
> >
> > Thanks for you response, my concerns are all addressed.

---

> ### Author Response · Authors · 2025-08-05
> **Reminder to Review Rebuttal**
>
> Dear Reviewer Pgyi,
>
> Thank you very much for your careful review of our manuscript during the initial review stage and **for your ​​overall positive and valuable feedback**​​. We greatly appreciate your ​​encouraging comments and constructive suggestions​​.
>
> We have addressed the points you raised in our rebuttal with detailed responses and clarifications. We hope these replies satisfactorily resolve your questions.
>
> As the deadline for the rebuttal discussion period is approaching quickly, we would be most grateful if you could find time to review our response. Should you have any further comments or suggestions, we would sincerely appreciate your continued feedback at your earliest convenience. Your insights are invaluable for further improving our work.
>
> Sincerely,
> Anonymous Authors

---

### Official Review · Reviewer_SDK1 · 2025-07-03

**Rating:** 2
**Confidence:** 3

**Summary:**

This paper introduces CellVerse, a language-centric benchmark designed to evaluate the capabilities of LLMs on a hierarchy of single-cell analysis tasks. Through a comprehensive evaluation, the work demonstrates that current models have a very limited understanding of cell biology and perform poorly on key reasoning tasks, often no better than random guessing, thereby highlighting a significant gap between the promise and reality of applying LLMs in this domain.

**Dataset Code Accessibility:**

Yes

**Ethical Considerations:**

No, there are no or only very minor ethics concerns

**Final Justification:**

Thank you for your comments and clarifications. I still find it difficult to be fully convinced by the significance and motivation of this study. The main contribution appears to be the use of LLMs to represent existing data rather than releasing new data, which limits its novelty.

**Limitations Weaknesses:**

- The benchmark is designed as an evaluation suite to assess current LLM capabilities. While effective for controlled initial assessment, this targeted design may limit the benchmark's broader applicability and longevity as a foundational resource.

- The benchmark is constructed by transforming several existing, publicly available datasets into a unified QA format. The work represents significant curation effort rather than the creation of fundamentally new data, which may weaken its contribution. While this language-centric transformation approach is novel, the core contribution remains an application of existing data assets.

**Strengths Contributions:**

- The authors conduct extensive experiments using their proposed CellVerse dataset, a unified benchmark that brings together four different multi-omics data types, to systematically assess how 14 open- and closed-source LLMs perform across three hierarchical single-cell analysis tasks.

- The experiments show that larger, general-purpose models exhibit basic reasoning capabilities and outperform smaller, specialized models that were specifically fine-tuned for single-cell analysis tasks. While this finding isn't entirely unexpected, it provides compelling empirical evidence that model scale remains a critical driver of performance, even in highly specialized domains like cell biology.

---

> ### Author Rebuttal · Authors · 2025-07-30
>
> We are truly grateful for the time you have taken to review our paper, and your insightful comments and support. Your positive feedback is incredibly encouraging for us! In the following response, we would like to address your concerns and provide additional clarification.
>
> > W1. The benchmark is designed as an evaluation suite to assess current LLM capabilities. While effective for controlled initial assessment, this targeted design may limit the benchmark's broader applicability and longevity as a foundational resource.
>
> A1. Thank you very much for your thoughtful comment. We fully understand the concern regarding the potential limitations in the broader applicability and longevity of a targeted evaluation benchmark. However, we believe that our benchmark is designed with **long-term usability** and **scalability** in mind, and its applicability will not be restricted over time for several reasons:
> - **Fully Open-Sourced and Reproducibility**
> We have completely open-sourced the dataset, evaluation scripts, and model interface code. This ensures that the community can readily reproduce our results and apply the benchmark to future models, even as the landscape of large language models (LLMs) continues to evolve.
> - **Model-Level Scalability**
> As new LLMs are continuously emerging, our evaluation pipeline is designed to be easily adaptable to any model. Researchers can simply plug in a new model and evaluate its understanding of cell biology using the same standardized process.
> - **Data-Level Scalability**
> Our benchmark supports seamless extension on the data side. By leveraging the C2S (Cell-to-Sentence) and GRN (Gene Regulatory Network) strategies described in the paper, future researchers can convert newly collected raw single-cell datasets into textual and QA formats. This allows the benchmark to grow in coverage and adapt to new biological tasks over time.
>
> In summary, we believe our benchmark offers both **immediate value** and **long-term potential** as a foundational resource for evaluating LLMs in the domain of cell biology. Its open design and extensibility mechanisms ensure its continued relevance as the field advances.
>
> We hope this addresses your concern, and we are grateful for the opportunity to clarify this in our revision.
>
>
> > W2. The benchmark is constructed by transforming several existing, publicly available datasets into a unified QA format. The work represents significant curation effort rather than the creation of fundamentally new data, which may weaken its contribution. While this language-centric transformation approach is novel, the core contribution remains an application of existing data assets.
>
> A2. We sincerely thank the reviewer for the insightful comments. We appreciate the concern regarding the reuse of existing datasets, and we would like to take this opportunity to clarify our contribution and the rationale behind our design choices.
>
> Our work does **not aim to introduce new biological datasets**, and we fully acknowledge that the benchmark is constructed by transforming several **existing, publicly available** datasets into a unified QA format. The reason for this is that **curating new biological datasets falls outside the scope of this paper**, especially given the **high cost and complexity** involved in experimental data collection in cell biology. Moreover, there already exists a **rich ecosystem of high-quality, widely adopted** datasets in the community, which we believe are suitable and sufficient for our research goals.
>
> Importantly, our **core contribution** lies not in data creation, but in proposing a **new language-centric paradigm** for understanding and reasoning about cell biology. This paradigm shift—using **natural language as the interface** between biological knowledge and computational models—offers a novel perspective that we believe has long-term implications for how LLMs can be applied in biological research.
>
> We would also like to point out that this design choice is **well aligned with prior works** in the AI4biology community. Existing studies [1,2,3] also **rely entirely on existing datasets** while introducing new task formulations or data transformations. These works have made meaningful contributions to their respective subfields without the need for new data collection.
>
> To summarize:
>
> - Our main contribution is the **language-based reframing** of biological tasks and the **exploration of a linguistic interface** for cell biology.
> - We **intentionally leverage existing datasets** to ground our benchmark in well-validated biological contexts.
> - This **approach is common and accepted** in the AI4biology community, and does not diminish the novelty of the proposed paradigm.
> - In future work, we **welcome and encourage** researchers to collect new biological data where appropriate, to further enrich the language-centric methodology.
>
> We hope this addresses the reviewer’s concerns and helps clarify the motivation and positioning of our work. Thank you again for your thoughtful feedback.
>
> **Reference**
>
> [1] Wu et al., Contextualizing Biological Perturbation Experiments through Language, ICLR, 2025
> [2] Levine et al., Cell2Sentence: Teaching Large Language Models the Language of Biology, ICML, 2024
> [3] Rizvi et al., Scaling Large Language Models for Next-Generation Single-Cell Analysis, bioRxiv, 2025
>
>
> In light of these responses, we hope we have addressed your concerns, and hope you will consider raising your score. If there are any additional concerns that we have not yet addressed, please do not hesitate to share them, and we will promptly attend to them.

---

> > ### Author Response · Authors · 2025-08-06
> > **Appreciation and Hope for Further Comments**
> >
> > **Dear Reviewer SDK1,**
> >
> > We sincerely thank you for dedicating your time and effort to reviewing our manuscript. We also appreciate you completing the “Mandatory Acknowledgement” and carefully considering our rebuttal.
> >
> > We hope that our responses have adequately addressed your concerns. If you have any remaining questions or would like further clarification on any point, we would be happy to elaborate. Your thoughtful feedback is greatly appreciated and will help us improve the quality of our work.
> >
> > **Thank you once again for your time, support, and consideration.** We would welcome any additional comments you may have.
> >
> > Best regards,
> > Anonymous Authors

---

> ### Author Response · Authors · 2025-08-05
> **Reminder to Review Rebuttal**
>
> Dear Reviewer SDK1,
>
> Thank you very much for your careful review of our manuscript and for your valuable feedback during the initial review stage. We greatly appreciate your professional insights.
>
> We have addressed your concerns in our rebuttal with detailed responses and clarifications. We hope these replies satisfactorily resolve your questions.
>
> Additionally, we would like to bring to your attention the responses we have provided to the other reviewers in the rebuttal. We are grateful that **all 3 other reviewers offered positive assessments of our work**​​ during this round. Specifically, ​​**Reviewer tCEs and Reviewer A5e3 explicitly confirmed that their concerns have been adequately addressed in our responses**.​
>
> ​​As the deadline for the rebuttal discussion period is approaching quickly, we would be most grateful if you could find time to review our response.​​ Should you have any further comments or suggestions, we would sincerely appreciate your continued feedback at your earliest convenience. Your insights are invaluable for further improving our work.
>
> Sincerely,
> Anonymous Authors

---

### Author Response · Authors · 2025-08-08
**Summary of Rebuttal&Discussion**

**Dear Reviewers, AC, SAC, and PC,**

Thank you for your careful reviews, constructive suggestions, and continued support throughout the rebuttal period. As the discussion window nears closure, we summarize our key contributions and responses below for your fair evaluation, particularly noting that we have not received further feedback from Reviewer **SDK1**.

---

### Key Contributions：

1.  **First Comprehensive Benchmark:** Introduces the first language-centric benchmark tailored for single-cell analysis, spanning **4 types** of single-cell multi-omics data and **3 hierarchical levels** of single-cell analysis tasks (cell-level, drug-level, gene-level).
2.  **Systematic Evaluation & Insights:** Presents a rigorous evaluation of 14 open/closed-source LLMs on this benchmark. Our in-depth analysis offers actionable insights and potential directions for future LLM applications in cell biology.

---

### Reviewer Recognition:

We are grateful for the reviewers' acknowledgment of the manuscript's strengths:

*   **Scope & Evaluation:** "**impressive scope and evaluation**" (tCEs), "**broad coverage**" (Pgyi, tCEs), "**extensive experiments and systematic evaluation**" (SDK1, tCEs, A5e3, Pgyi).
*   **Novelty & Impact:** "**novel benchmark**" (Pgyi, tCEs), "**meaningful exploration**" (SDK1, tCEs, A5e3), "**potential impact**" (Pgyi, tCEs, A5e3), "**motivated by real bottlenecks**" (Pgyi).
*   **Clarity & Transparency:** "**well-structured and well-written paper**" (tCEs, A5e3), "**transparent reporting**" (Pgyi).

---

### Status of Concerns:

*   **Reviewer A5e3:** Acknowledged that our detailed responses **resolved all concerns**.
*   **Reviewer Pgyi:** Acknowledged that our detailed responses **addressed all concerns**.
*   **Reviewer tCEs:** Confirmed **all concerns were addressed** and further endorsed the work, calling it "**a great benchmark work that can inspire the paradigm of next-generation single-cell analysis**".
*   **Reviewers SDK1:** We did not receive active engagement during the discussion period. Our comprehensive analyses and additional experiments provided during the rebuttal fully address the initial concerns.

---

Thank you for your time and consideration.

Sincerely,
The Authors of Manuscript 642

---

### Decision · Program_Chairs · 2025-09-18

**Decision:**

Accept (poster)

**Comment:**

This manuscript introduces CellVerse, a QA benchmark for single-cell analysis tasks. This is an important and timely topic. All reviews but one recommended acceptance. On the positive side, reviewers commended the comprehensive nature of the benchmark, the clarity of the writing, the thoroughness of the documentation. The review recommending rejection cited the fact that this benchmark is constructed from publicly available data and that it may not stand the test of time.  Overall, I do not find these concerns compelling, as it is still a significant contribution to collect public data into a useful benchmark. I am recommending acceptance.